# Phosphorylation of ULK1 affects autophagosome fusion and links chaperone-mediated autophagy to macroautophagy

Chenyao Wang[1,2], Huafei Wang[1], Deyi Zhang[1], Wenwen Luo[1], Ruilong Liu[3], Daqian Xu[4], Lei Diao[3], Lujian Liao[5] & Zhixue Liu [1,6]

The Unc-51 like autophagy activating kinase 1 (ULK1) complex plays a central role in the initiation stage of autophagy. However, the function of ULK1 in the late stage of autophagy is unknown. Here, we report that ULK1, a central kinase of the ULK1 complex involved in autophagy initiation, promotes autophagosome–lysosome fusion. PKCα phosphorylates ULK1 and prevents autolysosome formation. PKCα phosphorylation of ULK1 does not change its kinase activity; however, it decreases autophagosome–lysosome fusion by reducing the affinity of ULK1 for syntaxin 17 (STX17). Unphosphorylated ULK1 recruited STX17 and increased STX17's affinity towards synaptosomal-associated protein 29 (SNAP29). Additionally, phosphorylation of ULK1 enhances its interaction with heat shock cognate 70 kDa protein (HSC70) and increases its degradation through chaperone-mediated autophagy (CMA). Our study unearths a key mechanism underlying autolysosome formation, a process in which the kinase activity of PKCα plays an instrumental role, and reveals the significance of the mutual regulation of macroautophagy and CMA in maintaining the balance of autophagy.

[1] Key Laboratory of Nutrition and Metabolism, Institute for Nutritional Sciences, Shanghai Institutes for Biological Sciences, Chinese Academy of Sciences, University of Chinese Academy of Sciences, Shanghai 200031, China. [2] Department of Immunology, Lerner Research Institute, Cleveland Clinic, Cleveland, OH 44195, USA. [3] Institute of Biochemistry and Cell Biology, Shanghai Institutes for Biological Sciences, Chinese Academy of Sciences, University of Chinese Academy of Sciences, Shanghai 200031, China. [4] Department of Neuro-Oncology, The University of Texas MD Anderson Cancer Center, Houston, TX 77030, USA. [5] Shanghai Key Laboratory of Regulatory Biology, School of Life Sciences, East China Normal University, Shanghai 200062, China. [6] Center of Molecular and Translational Medicine, Georgia State University, Atlanta, GA 30302, USA. Correspondence and requests for materials should be addressed to Z.L. (email: zxliu@sibs.ac.cn)

Autophagy is a process of metabolic degradation at the cellular level in which organelles or portions of the cytosol are sequestered by double-membrane autophagosomes and fused with lysosomes for degradation. Autophagy plays a crucial role in a variety of biochemical processes, including cell survival, oxidative stress, removal of redundant organelles and proteins, and resistance to infection by pathogens[1].

The ULK1/ATG13/FIP200 complex initiates the occurrence of autophagy in mammalian cells. ULK1 is a serine/threonine-specific protein kinase. The ULK1 complex is made of autophagy-related protein 13 (ATG13), which contains the HORMA domain, the FIP200 (RB1CC1) scaffold, and autophagy-related protein 101 (ATG101)[2]. The ULK1 complex senses the nutrient status of cells to initiate or terminate autophagy. ULK1 can be phosphorylated by mammalian target of rapamycin complex 1 (mTORC1) or AMP-activated protein kinase (AMPK) to prevent or promote autophagy[3,4]. As a kinase, ULK1 also phosphorylates many autophagy-related targets, such as beclin1 (BECN1), vacuolar protein sorting 34 (VPS34), and ATG101, that are important for autophagy initiation[5].

Protein kinase C (PKC) is a family of serine/threonine protein kinases. PKCs are activated by increases in the concentration of diacylglycerol (DAG) or calcium ions ($Ca^{2+}$)[6]. Fifteen members of the PKC family in humans are divided into three subfamilies based on their second messenger requirements: conventional (or classical), novel or atypical. PKC regulates autophagy by phosphorylating microtubule-associated protein 1 A/1B-light chain 3 (LC3), a marker of autophagy[7].

Autophagosome fusion into lysosomes degrades molecular components necessary for cell survival. Several SNARE (soluble N-ethylmaleimide-sensitive fusion (NSF) attachment protein receptors) proteins are involved in this step. They are STX17, vesicle-associated membrane protein 8 (VAMP8) and SNAP29[8]. Upon membrane fusion, the SNARE complex forms a parallel four-helix bundle consisting of the Qa-, Qb-, Qc-, and R-SNAREs. STX17 is a Qa-SNARE on autophagosomes that interacts with the Qb/Qc motif of SNAP29. This complex fuses with the R-SNARE named VAMP8, which is on lysosomal membranes, thus completing the fusion of autophagosomes and lysosomes. Autophagy-related 14 homolog (ATG14L) also plays a critical role in SNARE-mediated fusion[9]. It was previously reported that ATG14L also promotes autophagy in the initiating step via binding to the BECN1/VPS34/VPS15 complex[10], which suggests that one factor can play multiple roles in different stages of autophagy.

In the present study, we found that ULK1 mediates fusion of the autophagosome to the lysosome by interacting with STX17. This activity is dependent on its phosphorylation state via PKCα. When the nutrient signal activates PKCα, ULK1 is phosphorylated and degraded by chaperone-mediated autophagy (CMA).

## Results

### ULK1 is phosphorylated by PKCα at S423 in vivo and in vitro.

The ULK1/ATG13 complex has a central role in starvation-induced autophagy. ULK1 senses upstream signals from mTOR or AMPK, which can lead to prevention or activation of the downstream autophagy pathway. PKC inhibits autophagy by directly phosphorylating the central conjugation system protein LC3. We speculated that PKC may phosphorylate other autophagy-related proteins to regulate autophagy. Therefore, we used a p-PKC-substrate antibody to determine the phosphorylation status of key autophagy-related proteins. These proteins include Vps34, BECN1, ULK1, ATG14L, and autophagy-related protein 7 (ATG7). In our assay, only ULK1 showed a clear signal (Fig. 1a). The expression of these proteins was determined by

western blot with indicated antibodies (Supplementary Figure 1a). The p-PKC-substrate antibody recognizes cellular proteins only when phosphorylated at Ser residues surrounded by Lys or Arg at the +2 and −2 positions along with a hydrophobic residue at the +1 position. To pin-point the specific phosphorylation site, ULK1 was truncated to 4 GST-tagged fragments and transfected into HEK293T cells with PKC and subsequent detection of the pull down by p-PKC-substrate antibody. A strong band was detected in the fragment 279–525, indicating that the specific phosphorylation site is located on fragment 279–525 (Fig. 1b). LC-MS/MS was performed to precisely identify phosphorylation sites after the in vitro PKCα kinase assay. (Supplementary Figure 1b). The molecular weight of the precursor ion indicates that the peptide is singly phosphorylated. The existence of unphosphorylated y15 ions suggests that the phosphorylation site is located in one of the first two serine residues (S422 and S423), although a lack of b ions made localization of the phosphorylation site uncertain. The potential PKC phosphorylation sites on fragment 279–525 were predicted by GPS software. Thus, alanine scanning was used to identify the specific phosphorylation site by an in vitro kinase assay with purified PKCα from insect cells. We identified that S423 was the phosphorylation site of ULK1 by PKCα (Fig. 1c). This phosphorylation site is conserved from Xenopus tropicalis to Homo sapiens (Fig. 1d). To confirm the phosphorylation site, we made a specific antibody to recognize the phosphorylation of S423. WB with this specific phosphorylation antibody was used in Myc-ULK1 WT and S423A expressing HEK293T cells. The result showed a strong band in ULK1 WT but no signal in S423A (Fig. 1e), indicating that this antibody specifically recognized the ULK1 S423 phosphorylation. To further verify this phosphorylation, we performed an in vitro kinase assay. GST-ULK1 WT and S423A were transfected into HEK293T cells, and the pull-downs were performed in the presence and absence of CIP (Alkaline Phosphatase, Calf Intestinal). The kinase assay was performed and detected with the p-S423 antibody. While ULK1 WT displayed a clear band, S423A did not (Fig. 1f). To confirm that PKCα was the upstream kinase for ULK1, we cotransfected two types of PKCα (PKCα-DN, dominant negative; PKCα-WT, wild type) into HEK293T. ULK1 was IPed and analyzed with the p-S423 antibody. The panel of PKCα-WT transfections was detected by a strong signal using the p-S423 antibody, but very faint signals were observed in PKCα-DN transfections (Fig. 1g). This may due to the endogenous PKCα in the PKCα-DN transfections cells. From the Human Protein Atlas database, we found that PKCα is dominant in U87 cells (Supplementary Figure 1c). To confirm the upstream kinase of S423, we alternated PKCα activity with activators and inhibitors, including a PKCα knockdown in U87 cells with a pGIPZ lentivirus. Both knockdown of PKCα and inhibition of the activity of PKC reduced the phosphorylation level of S423 (Fig. 1h, i).

Generally, kinases interact with substrates by transferring a phosphate to a hydroxyl group. The interaction provides indirect evidence for the kinase and substrate[11]. We speculated that PKCα and ULK1 could bind to each other. To answer this question, we transfected HA-PKCα with GST-ULK1 or GST only. The pull-down assay showed that ULK1 interacts with PKCα (Supplementary Figure 2a). We also performed an IP assay using an anti-ULK1 antibody or IgG to IP endogenous PKCα from HeLa cell lysate. The result demonstrated that only the anti-ULK1 antibody could IP PKCα (Supplementary Figure 2b). Exogenous and endogenous binding assays showed that ULK1 interacts with PKCα (Supplementary Figure 2a, b). This interaction was confirmed by an in vitro binding assay using recombinant GST-ULK1 279–525 (from bacteria) and His-PKCα (from insects) (Supplementary Figure 2c). These results indicated that PKCα interacts with ULK1 and is a kinase for ULK1. Interestingly,

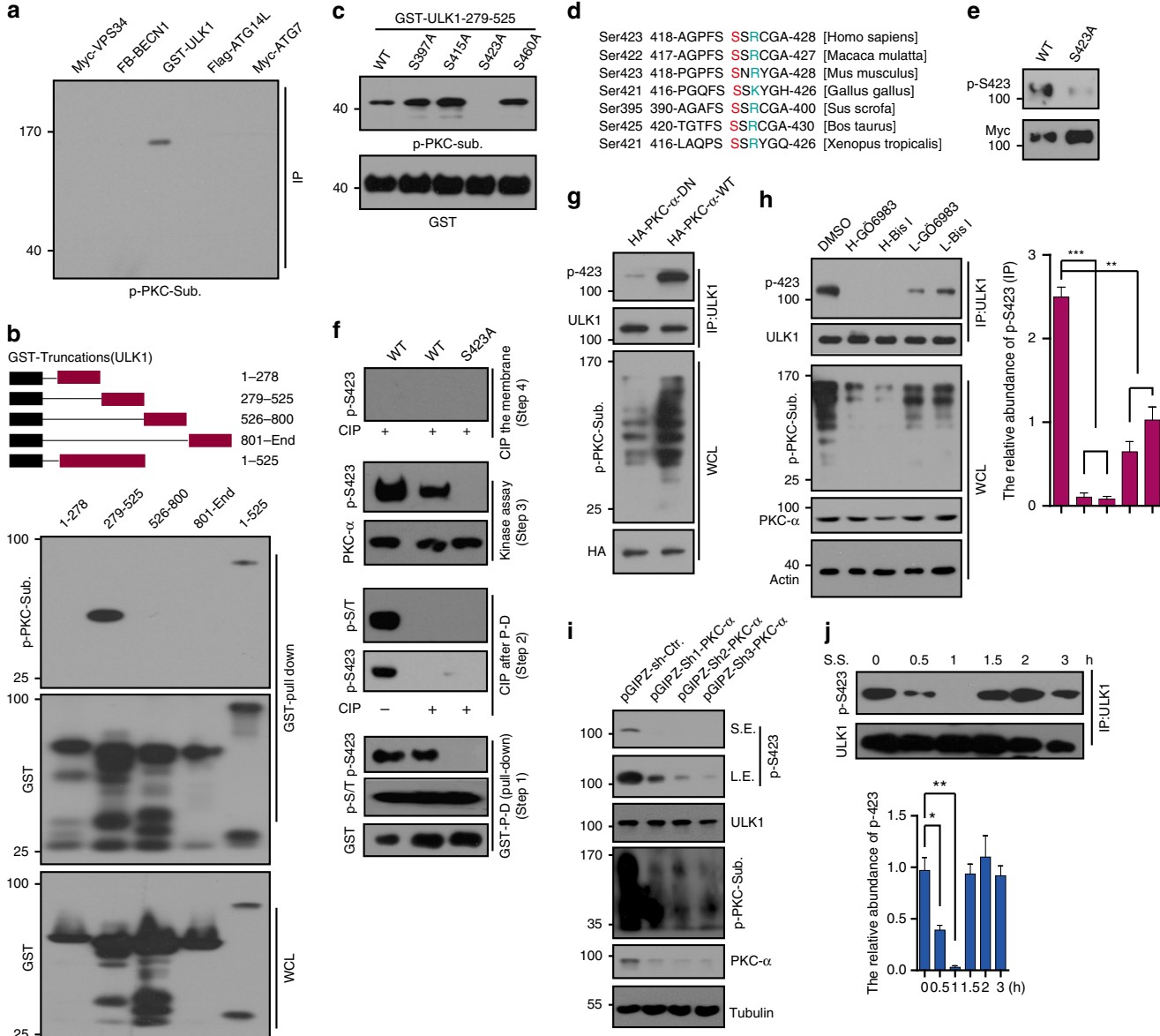

**Fig. 1** S423 of ULK1 is the phosphorylation site by PKCα. **a** Phosphorylation of autophagy-related proteins was detected using p-PKC-Substrate antibody. The expression of these proteins is shown in Supplementary Figure 1a. **b** GST-ULK1 truncations were transfected and the cell lysates were pulled down for WB analysis with p-PKC-substrate antibody. The phosphorylation site is between 279–525AA. **c** Mapping phosphorylation site of ULK1 by in vitro kinase assay. S423 is the phosphorylation site. **d** Evolutionary conservation of the ULK1 Ser423. **e** We generated a p-S423 antibody by ABclonal Technology, and verified the p-S423 antibody by WB. **f** Phosphorylation site was confirmed by in vitro kinase assay with full-length ULK1. The indicated GST-ULK1s were transfected into HEK293T cells, followed by GST pull down (Step1), Dephosphorylation of the indicated beads by CIP (Step2), PKCα kinase assay (Step3), and CIP treated the whole PVDF membrane to confirm the phosphorylation (Step4). **g** The phosphorylation of endogenous ULK1 was detected by p-S423 antibody after HEK293T cells were transfected with PKCα WT or DN (dominant negative). **h** p-ULK1 S423 was analyzed by WB after U87 cells were treated with PKC inhibitors GÖ6983 or Bis I for 0.5 h. p-PKC-substrate antibody was used to determine the efficiency of drugs. Actin, ULK1, and PKCα were used as loading controls. The relative abundance of p-S423 was quantified and shown in the right panel. **i** U87 cells were infected with lentivirus of non-targeting or shPKCαs for 48 h. The lysates were analyzed with p-S423 antibody by WB. Quantification of p-S423. The efficiencies of shPKCαs were analyzed by PKCα antibody. Actin was used as loading controls. **j** p-S423 was determined during serum starvation by WB. p-S423 was consistent with PKCα activity. p-S423 was quantified and is shown in the bottom panel. Statistical significance was measured via unpaired and two-tailed Student's t-tests and is presented as follows: *$p < 0.05$, **$p < 0.01$, and ***$p < 0.001$. All error bars indicate SEM. Bars are mean ± SEM of triplicate samples. S.S.: serum starvation, WCL: whole-cell lysate, S.E.: short exposure, L.E.: long exposure

p-S423 of ULK1 and its mimic S423D interact with PKCα with a higher affinity than ULK1 S423A (Supplementary Figure 2d, e). To investigate the specific domains contributing to the interaction between ULK1 and PKCα, we truncated PKCα to 7 fragments (GST tagged) and cotransfected with the Flag-ULK1 into HEK293T cells. IP result showed that the 167–338 region

(fragment 2) of PKCα is responsible for ULK1 binding (Supplementary Figure 2f). Fragments 4 and 5, which only contain a portion of fragment 2, do not bind ULK1. We surmised that the reason this occurred is because the conformation for ULK1 binding requires a whole fragment 2. By a similar method, we determined that the 279–525 region of ULK1 interacts with

PKCα (Supplementary Figure 2g), which is consistent with this portion being the phosphorylation region.

It is widely documented that when serum starvation occurs, the phosphorylation states of many autophagy-related proteins become altered in response to upstream kinases change. This change has major physiological significance. It is well known that autophagy increases due to serum starvation. The first peak of autophagy occurs at 1 h of serum starvation, then it goes down at approximately 1.5 h[12], followed by a sharp increase after 3–4 h[12,13]. The autophagy level appears as a wavepeak over the serum starvation time course. Correspondingly, PKC activity shows a concave shape during serum starvation (Supplementary Figure 2h), which strongly hints at a correlation between PKC and the phosphorylation of autophagy genes. To explore whether this phosphorylation of ULK1 is related to autophagy, we performed a time course of serum starvation in HeLa cells. We determined the phosphorylation of ULK1 with the p-S423 antibody. The result showed that this phosphorylation is related to autophagy. p-ULK1 was quantified (Fig. 1j). To confirm the role of p-ULK1 S423 in autophagy, we performed experiments to detect autophagy in the livers of mice with chew diet or 16 h of fasting. The liver lysate was analyzed by p-PKC-substrate, LC3, and p-S423 antibodies. With fasting, the autophagy level strongly increased, but PKC activity decreased along with p-S423 (Supplementary Figure 2i). Thus, p-S423 may play a crucial role in starvation-induced autophagy. Taken together, PKCα physically interacts and phosphorylates ULK1 at S423. This phosphorylation may be involved in autophagy regulation.

**Phosphorylation does not affect early stage of autophagy.** The core ULK1 complex contains ULK1, FIP200, and ATG13, which are essential for initiation of the autophagy process[14]. To explore whether this phosphorylation influences the ULK1 complex, we transfected Myc-ULK1 WT, S423A, S423D with Flag-ATG13 or HA-FIP200 into HEK293T cells and performed an IP assay with an anti-Myc antibody or IgG as the negative control. The results showed that there were no affinity differences between ULK1 variants and ATG13 or FIP200. There were also no significant quantitative differences (Fig. 2a, b).

Enhancement of the interaction and phosphorylation of Beclin1 via ULK1 increases the level of autophagy[8]. To analyze whether the phosphorylation of ULK1 affects the interaction between ULK1 and BECN1, we performed an IP assay and found no change in the binding of Beclin1 with ULK1 variants (Fig. 2c). Similar experimental results were obtained by a semi-endogenous protein IP assay. There were no significant differences in ULK1 variants binding ATG13, FIP200, or BECN1 (Fig. 2d).

As a kinase, ULK1 promotes autophagy by phosphorylating BECN1 and ATG13. To further explore whether this phosphorylation affects its kinase activity, we transfected ULK1 WT, S423A, and S423D into HeLa cells. The kinase activities were detected by p-ATG13 and p-BECN1. The results showed that this phosphorylation does not affect ULK1 activity or PKCα and mTORC1 activities (Fig. 2e).

We next assessed the impact of PKC inhibition on ULK1 kinase activity. After treatment with different doses of the PKC inhibitor GÖ6983, we determined the level of p-Atg13 S318 and p-Beclin1-S14 in HeLa cells. We found that the phosphorylation of p-Atg13 S318 increased, indicating that ULK1 kinase activity increased. These results may be due to a reduction in mTORC1 activity because these inhibitors decreased mTORC1 activity but did not affect AMPK (Fig. 2f, g). Overall, PKCα-mediated phosphorylation of ULK1 did not affect the initiation of autophagy.

**Phosphorylation of ULK1 regulates autolysosome formation.** Active PKC significantly attenuates autophagy. We used the PKC inhibitors to induce autophagy by analyzing LC3 and SQSTM1 (P62), both are markers of autophagy. To explore whether PKC inhibition induced autophagy is mediated by mTORC1 activity, we combined PKC and mTORC1 inhibitors in HeLa cells. Inhibition of both PKC and mTORC1 led much stronger autophagy than inhibition of only mTORC1 or PKC (Supplementary Figure 3a). When the individual inhibitors of PKC or mTORC1 induced autophagy were compared, Rapamycin with chloroquine (CQ) caused the most LC3-II accumulation in U87 cells of all tested inhibitors (Supplementary Figure 3b). To better understand the relationship between mTORC1 and PKCα, we knocked down Raptor or mTOR in HEK293 and HeLa cells[15] or PKCα in both cell lines. We measured PKC and mTORC1 activities upon knockdown of Raptor or mTOR, which decreased both mTORC1 and PKC activities. Knockdown PKCα also led to reductions in PKC and mTORC1 activities (Supplementary Figure 3c, d, e). These results indicate that PKCα and mTORC1 affect each other.

To analyze the role of phosphorylation of ULK1 by PKCα in autophagy, we first knocked out ULK1 and then knocked down ULK2 (ULK1/2-KO/D) in HeLa cells. Then, we transfected Myc-ULK1 WT (wild type), S423A (an unphosphorylated mutant) and S423D (a phosphorylated mimic) into ULK1/2-KO/D HeLa cells. We determined the autophagy markers LC3 and SQSTM1 after ULK1/2-KO/D HeLa cells were treated with or without the lysosome inhibitor chloroquine. We obtained an unexpected result. The abundance of LC3-II in S423D was higher than that in S423A and WT in normal conditions. With CQ treatment, the accumulation of LC3-II in CTRL, S423A, and WT significantly increased. However, the accumulation of LC3-II increased much less in S423D cells (Fig. 3a). We normalized the abundance of LC3-II and P62 based on actin, and the ratios of P62 and LC3-II abundance between the CQ and DMSO groups were calculated. The abundance of LC3-II increased more than 3-fold and P62 increased 4-fold in the S423A panel when CQ was added, whereas the increases in the S423D group were only 1.2- and 1.9-fold for LC3-II and P62, respectively (Fig. 3b). P62 accumulated in both CQ-treated and untreated ULK1/2-KO/D cells. No LC3-II was detected in ULK1/2-KO/D cells. We observed similar results in ULK1 variants overexpressed in HeLa cells (Supplementary Figure 4a). To confirm this result, we performed immunostaining of LC3 in the transfected cells. The representative images and graphs of statistical analysis are shown in Fig. 3 (Fig. 3c, d). Much more puncta were found in S423D overexpressed cells than in WT and S423A overexpressed cells under normal conditions. CQ treatment leads to a puncta number increase eight times that of WT and S423A overexpressed cells (Fig. 3d). However, the number of puncta is not significantly increased in S423D overexpressed cells (Fig. 3c). The results were also confirmed by transmission electron microscopy (TEM). Many autophagosomes and autolysosomes were observed in wild-type cells, but they were seldom observed in ULK1/2-KO/D cells (Supplementary Figure 4b). In ULK1/2 KO/D cells, we expressed the ULK1 variants stably and performed examination by TEM. We observed more autophagosomes and autolysosomes in S423D transfected cells than in WT and S423A transfected cells. These structures did not substantially increase after CQ treatment in S423D cells. However, they significantly increased in WT and S423A cells with CQ treatment (Supplementary Figure 4c). To confirm the role of ULK1 phosphorylation in membrane fusion, we performed an in vitro fusion assay. The autophagosomes (red) were preincubated with purified GST-ULK1 WT, S423A, or S423D for 10 min, then incubated with lysosomes (green) for 60 min. The reaction mixtures were fixed and mounted on glass coverslips for imaging with a confocal microscope (Fig. 3e).

GST-ULK1 S423D retarded the fusion of autophagosome to lysosome. In summary, phosphorylation of ULK1 regulates autophagosome and lysosome fusion.

**ULK1 regulates autophagy fusion through STX17.** Phosphorylation of ULK1 plays a role in autophagosome–lysosome fusion. However, the mechanism is unknown. It has been reported that

ULK1 and LC3 colocalize when autophagy occurs[16]. This colocalization strongly suggests that ULK1 plays a role in the late stages of autophagy. It was also reported that STX17 mediates the fusion of autophagosomes to lysosomes[17]. We speculated whether ULK1 could interact with STX17. To detect this interaction, we transfected Myc-tagged ULK1 and Flag-SBP tagged STX17 into HEK293T using anti-Myc antibody for immunoprecipitation (IP) and found that ULK1 binds to STX17 (Fig. 4a). We next

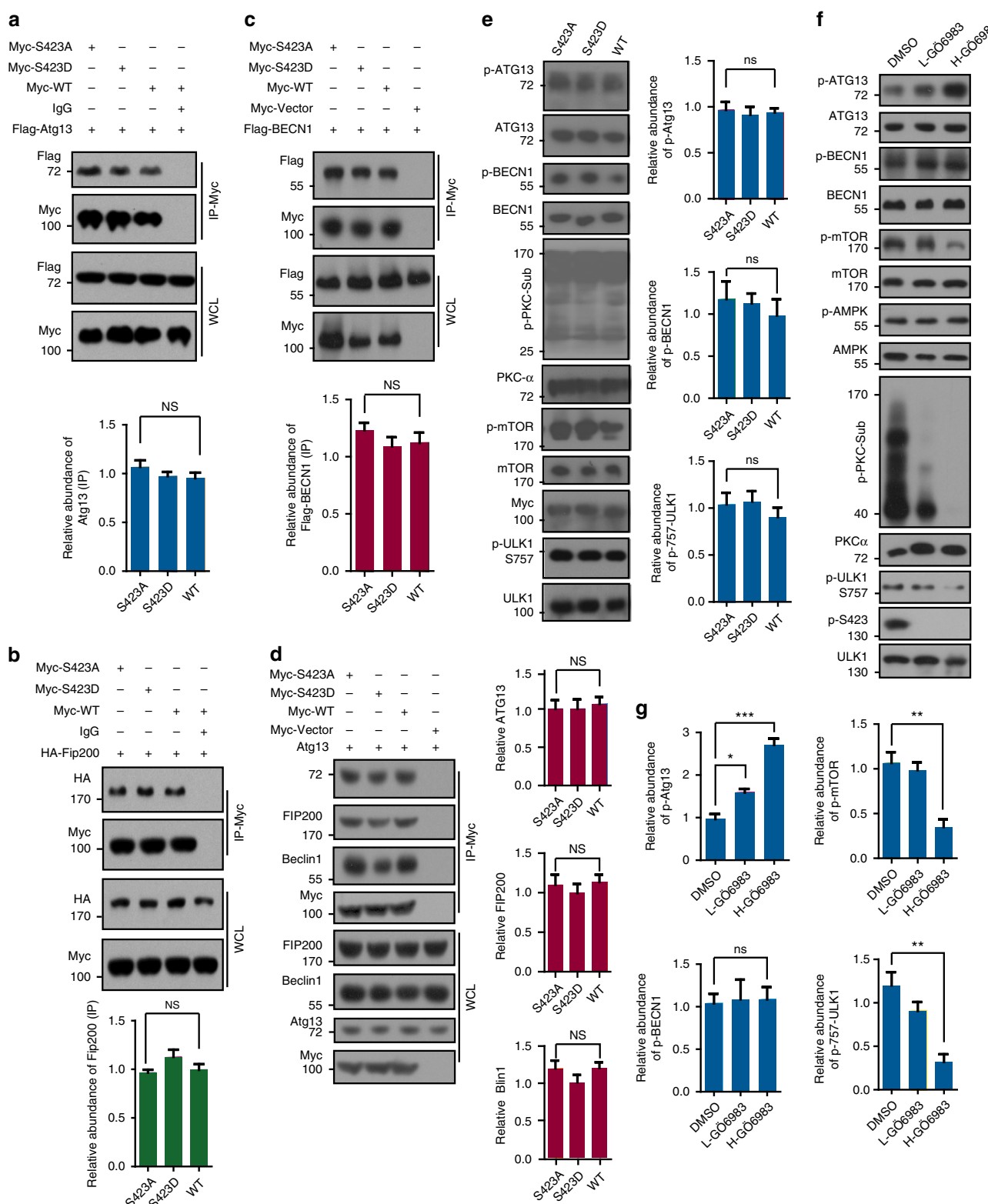

performed co-IP experiments with an anti-STX17 antibody and an anti-ULK1 antibody to assess endogenous protein interactions and confirmed that STX17 and ULK1 bind to each other in vivo (Fig. 4b, c). We also used mass spectrometry to analyze the GFP-STX17 IPed proteins, and the results confirmed ULK1 and STX17 binding (Supplementary Figure 5a, b). To determine which domains are responsible for STX17 and ULK1 binding to each other, GFP-tagged STX17 was truncated into six fragments. Through the co-IP experiment, we found that amino acids 157–275 of STX17 represent the domain that binds to ULK1 (Fig. 4d). With a similar strategy, we determined that the 279–525 domain of ULK1 binds to STX17 (Fig. 4e). We confirmed the interaction by an in vitro pull-down assay with recombinant proteins of GST-ULK1 279–525 and His-STX17 157–275. GST-ULK1 279–525 and His-STX17 157–275 were purified from bacteria and incubated in IP lysis buffer with glutathione beads for 4 h at 4 °C. Then, the beads were washed and analyzed by western blot (Fig. 4f). This interaction was also confirmed by PLA technology. GFP-STX17 was cotransfected with Myc-ULK1 or Myc vector into HeLa cells. Anti-Myc and anti-GFP antibodies were used for PLA assay. The red signal was detected in Myc-ULK1 transfected cells (Fig. 4g). These data show that ULK1 physically interacts with STX17.

Autophagy displays a waveshape over a time course of serum starvation[12]. To explore whether the affinity of ULK1 and STX17 displays a similar pattern during serum starvation, we performed a co-IP assay. We found that this interaction showed a peak at 1 h of serum starvation, followed by a decrease (Fig. 5a). This result is consistent with p-ULK1 (Fig. 1j). This interaction was confirmed by co-IP of Myc-ULK1 S423A, WT, and S423D with STX17. Phosphorylated mimic S423D bound less STX17 (Fig. 5b). Myc-ULK1 was transfected into HEK293 cells, and IP was performed. The beads were equally separated into three tubes, followed by CIP or PKCα kinase assay treatment. After incubation with the same amount of normal cell lysate and washing, endogenous STX17 was detected. We found that CIP treated ULK1 increased binding, but kinase treated ULK1 decreased the binding (Supplementary Figure 5c).

To study the interaction of these two proteins in vivo, mice were injected with PBS or PKC inhibitor GÖ6983 for 7 days by tail vein. Liver lysate was IPed with an anti-STX17 antibody. ULK1 was detected in IPs. We found that GÖ6983 treatment decreased p-S423 of ULK1 and enhanced affinity towards STX17 (Fig. 4h). GFP-STX17 was transfected into HeLa cells on slides, and serum starvation was performed. A PLA assay was carried out, and serum starvation increased the red signal, reflecting the interaction of ULK1 and STX17. The signal was quantified, and serum starvation treatment increased 2.5-fold (Fig. 4i). GFP-STX17 was cotransfected with Myc-ULK1 WT, S423A, S423D, or Myc vector into HeLa cells on slides. A PLA assay was performed,

and the results showed that unphosphorylated mimic ULK1 S423A significantly enhanced the red signal (Fig. 4j). We transfected Myc-ULK1 WT, S423A, S423D with GFP-SXT17, and Flag-LC3 into HeLa cells and performed immunostaining. The images showed that most SXT17 was colocalized with LC3 as puncta. Some ULK1 WT and S423A formed puncta and colocalized with STX17 and LC3, but this colocalization did not occur when S423D was used (Fig. 5c). The quantitative analysis is shown in Fig. 5d. We also performed immunostaining with ATG13, ATG16L1, and LAMP2. Similar to the above result, all ULK1 variants colocalized with ATG13. This result is consistent with previous data showing that phosphorylation of ULK1 does not affect autophagy initiation. However, quantitative analysis showed that the puncta of ULK1 colocalized with STX17 in WT and S423A, but STX17 did not colocalize with ATG13 (Supplementary Figure 6a, d). ATG16L1 is located in a prestructure of the autophagosome. We found that all ULK1 variants colocalized with ATG16L1. It is consistent with our observation that the phosphorylation does not affect ULK1 Kinase activity and autophagy flux. Partial STX17 colocalized with ATG16L1 (Supplementary Figure 6b). Quantitative analysis of the colocalization of ULK1 and ATG16L1 is shown in Supplementary Figure 6e. Lamp2 is a membrane protein of lysosomes and is used as a lysosome marker. All ULK1 variants colocalized with LAMP2. STX17 also colocalized with LAMP2, although less colocalization was observed in S423D transfected cells (Supplementary Figure 6c). The colocalization of ULK1 and LAMP2 was quantitatively analyzed (Supplementary Figure 6f). S423A, which promotes autophagy, does not display more colocalization with LAMP2, suggesting that S423A may be dissociated from the autophagosome after recruiting STX17. It was surprising that S423D had comparatively more colocalization with LAMP2.

We wanted to determine the mechanism that regulates fusion of the autophagosome to the lysosome by phosphorylation of ULK1. Thus, we transfected Flag-SBP-STX17 and GFP-Snap29 with variants of GST-ULK1 into HEK293T cells. We performed an IP assay with SBP beads and detected GFP-SNAP29 with/out CQ treatment. In normal conditions, STX17 bound more SNAP29 in ULK1 S423A cells than in the WT and S423D counterparts (Fig. 5e), indicating that ULK1 S423A increases fusion of the autophagosome to the lysosome through increasing affinity of SXT17 to SNAP29. We also observed increased binding of STX17 to SNAP29 in ULK1 WT and S423A transfection with CQ treatment (Fig. 5e). We next explored the interaction between STX17 and VAMP8 under similar conditions, and the results showed that ULK1 S423A had a stronger ability to bind VAMP8 with bafilomycin A (a lysosome inhibitor) treatment (Fig. 5f). Myc-ULK1 WT, S423A, S423D, or Myc vectors were transfected into HeLa cells in the presence and absence of CQ treatment.

**Fig. 2** Phosphorylation of ULK1 does not affect the ULLk1 and ATG13 complex and its kinase activity. **a** Flag-ATG13 was transfected with Myc-ULK1 WT, S423A and S423D into HEK293T cells for 24 h. The IPs were analyzed with Myc antibody. The densities of ATG13 were quantified with image J and showed in the bottom panel. **b** HA-FIP200 was transfected with Flag-ULK1 WT, S423A, and S423D into HEK293T cells for 24 h. IP was performed with Myc antibody and analyzed with HA antibody. The densities of Fip200 were quantified and showed in the bottom panel. **c** Flag-BECN1 was transfected with Myc-ULK1 WT, S423A, and S423D or Myc-Control into HEK293T cells for 24 h. IP was performed with Myc antibody and analyzed with Flag antibody. The densities of BECN1 were quantified and showed in the bottom panel. **d** IP of ULK1 complex partners by Myc-ULK1 variants transfected into HEK293T cells for 24 h was determined with ATG13, Fip200, BECN1 antibodies. The densities of ATG13, Fip200, and BECN1 are quantified and showed in the right panel. **e** ULK1, PKCα and mTORC1 activities were analyzed by WB after Myc-ULK1 WT and mutants were transfected into HeLa cells for 24 h. The densities of p-ATG13, p-BECN1, and p-ULK1 S757 are quantified and showed in the right panel. **f** HeLa cells were treated with different concentrations of PKC inhibitor GÖ6983 for 4 h. Lysates were analyzed with the indicated antibodies. **g** The densities of p-ATG13, p-BECN1, p-ULK1 S757, and p-mTORC1 were quantified from **f**. Statistical significance was measured via unpaired and two-tailed Student's $t$-tests and is presented as follows: NS: no significance; *$p < 0.05$, **$p < 0.01$, and ***$p < 0.001$. All error bars indicate SEM. Bars indicate mean ± SEM of triplicate samples. L-GÖ6983: low concentration, IC50; H-GÖ6983: high concentration, 5*IC50. WCL: whole-cell lysate

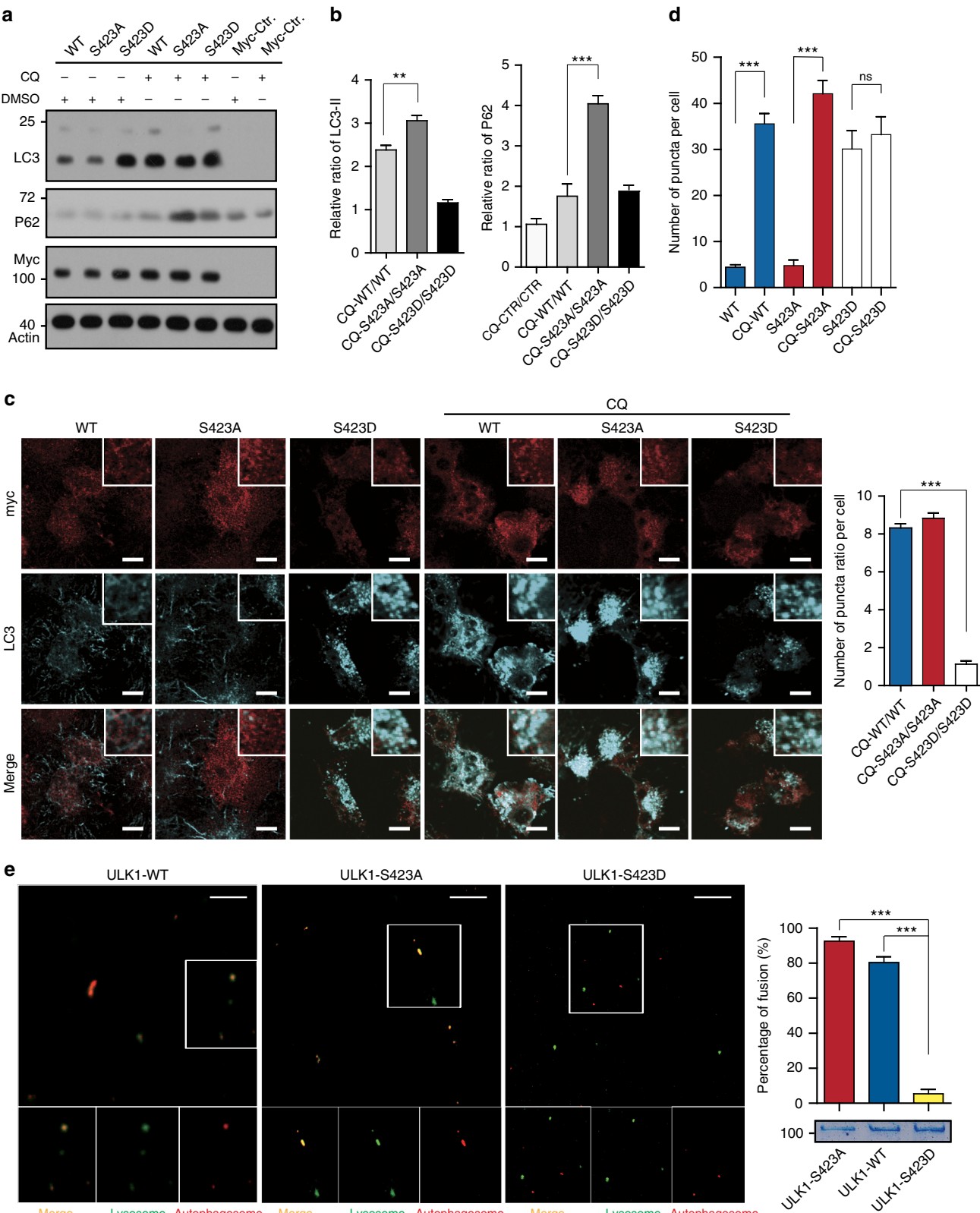

**Fig. 3** Phosphorylation of ULK1 plays a key role in fusion of autophagosomes to lysosomes. **a** The control vector, ULK1 WT, S423A, and S423D were transfected into the *ULK1/2-KO/D* cells and treated with/out Chloroquine (CQ, an lysosomal inhibitor, 20 nM). The lysate was analyzed by WB with LC3, p62 and actin antibodies. **b** The quantified ratio (CQ-ULK1s/ULK1s) of LC3-Ⅱ and p62 is shown. **c** Representative confocal images of *ULK1/2-KO/D* HeLa cells transfected with Myc-ULK1 WT and mutants treated with/out CQ shown as the average of puncta per cell. Scale bar, 10 µm. A minimum of 20 cells were counted. **d** The quantified ratio (CQ-ULK1s/ULK1s) of puncta per cell is shown. **e** Representative confocal images of in vitro fusion assay. Lysosomes (green) and autophagosomes (red) are mixed with purified GST-ULK1 WT, S423A, and S423D. Scale bar, 5 µm. The fusion percentage was quantified by comparison of the number of yellow dots to the number of red dots and yellow dots. All values are means ± SEM of three independent experiments. Student's *t* test (unpaired); NS: no significance; *$p < 0.05$, **$p < 0.01$, and ***$p < 0.001$

The anti-STX17 antibody or IgG was used for an IP assay. The IPs were analyzed with anti-Myc, anti-SNAP29, and anti-VAMP8 antibodies. STX17 bound more SNAP29 and VAMP8 in S423A expressing cells than in WT or S423D. CQ treatment increased the presence of the STX17/SNAP29/VAMP complex in all transfections (Fig. 5g). Our results were verified by a co-IP assay using mice liver lysates. The mice fasted for 16 h or were fed a chow diet, and the liver lysates were used for an IP assay with an

STX17 antibody. STX17 bound more ULK1 in fasting livers (Fig. 5h).

We further explored the mechanism of how ULK1 phosphorylation affects the STX17–SNAP29–VAMP8 complex. We found that overexpression of SNAP29 decreased the interaction of ULK1 and STX17 and that knockdown of *SNAP29* increased their interaction (Supplementary Figure 7a, b). However, knockdown of VAMP8 did not affect the interaction of ULK1 and STX17

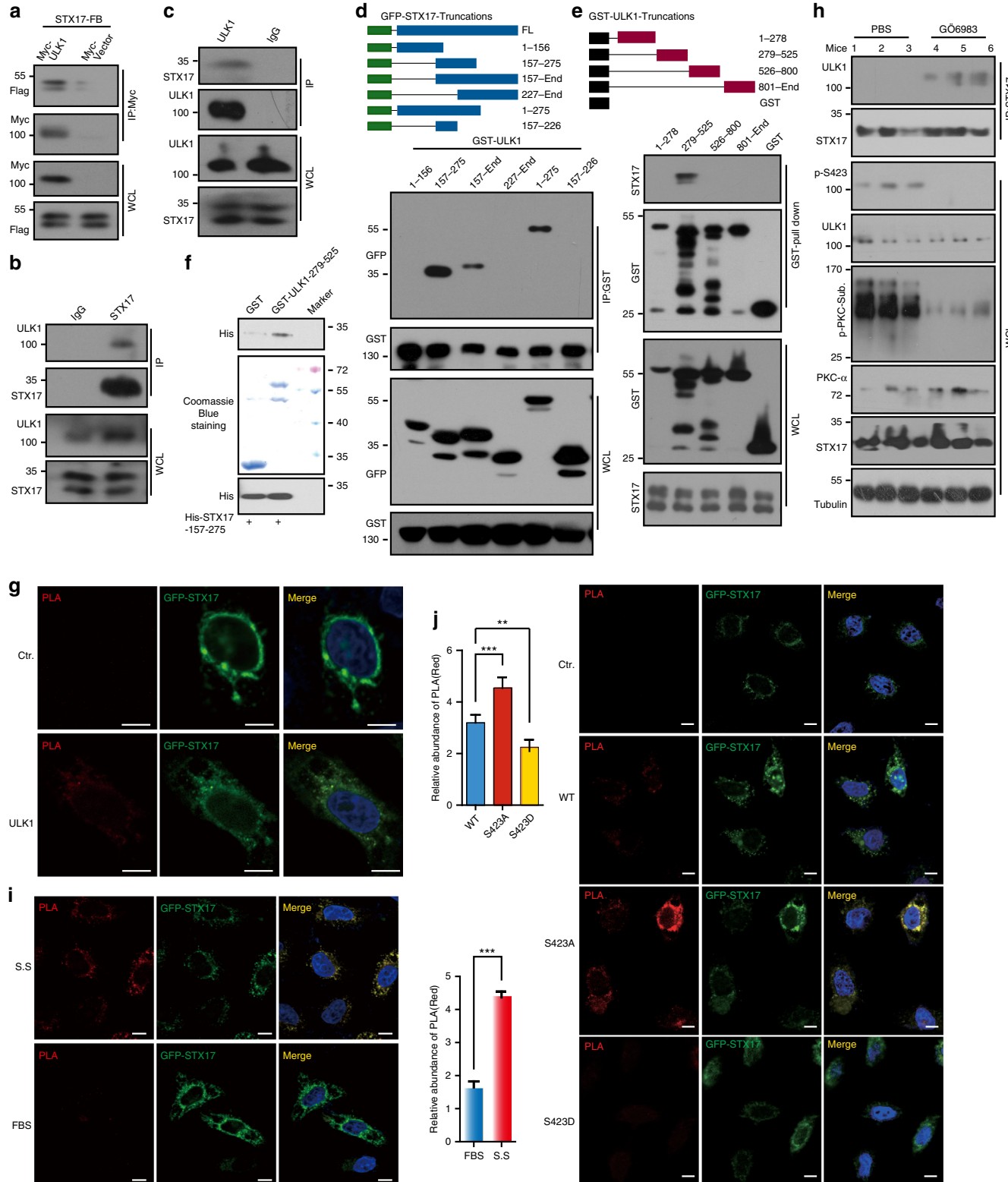

(Supplementary Figure 7c). These data indicated that unphosphorylated ULK1 recruits STX17 to autophagosomes and that STX17 has a greater affinity for SNAP29 when it is located on the surface of autophagosomes. Thus, STX17 binds to SNAP29 and releases ULK1 into the cytosol. These data demonstrate that ULK1 phosphorylation regulates the fusion of autophagosomes to lysosomes by regulating STX17–SNAP29–VAMP8 complex formation.

**CMA is a counterbalance for macroautophagy**. From the above results, we know that ULK1 can regulate autophagy via its phosphorylation by PKCα. To analyze whether phosphorylation affects the stability of ULK1, we constructed doxycycline-inducible Flag-ULK1 WT, S423A, and S423D variants. We transfected these plasmids into HEK293T cells and induced ULK1 expression with doxycycline, pretreated with cycloheximide (CHX, an inhibitor of protein synthesis), and performed MG132 incubation. MG132 inhibits proteasome-mediated protein degradation. MG132 prevents S423A degradation but not WT or S423D degradation (Supplementary Figures 8a, 6a), indicating that degradation of S423A occurs through the proteasome. The lysosome is another major protein degradation organelle. To explore whether ULK1 S423D is degraded through the lysosome, we used lysosome inhibitor chloroquine (CQ) to perform the experiment. We transfected HEK293T cells with Flag-ULK1 variants, pretreated cells with CHX for 3 h and then added CQ in the presence of CHX. We found that CQ inhibited ULK1 S423D degradation but not degradation of WT or S423A (Supplementary Figures 8b, 6b). ULK1 WT was degraded in both MG132 and CQ treatments because ULK1 WT could switch between the phosphorylated and unphosphorylated forms. These two forms are balanced within the cell. We observed that simultaneous usage of MG132 and CQ on Flag-ULK1 WT transfected HEK293 cells blocked ULK1 degradation (Supplementary Figure 8c). These results, combined with previous data showing that unphosphorylated ULK1 promoted autophagosome and lysosome fusion, might explain why all ULK1 variants colocalize with LAMP2. To further analyze whether ULK1 could be degraded through the lysosome, we knocked out Lamp2 with the CRISPR-Cas9 system. In *Lamp2 KO* cells, ULK1 accumulated (Fig. 6c). Lamp2a plays a critical role in chaperone-mediated autophagy. In addition, Lamp2a is the dominant form in HeLa and HEK293T cells. Therefore, we speculated that ULK1 may be degraded through CMA. To address this possibility, we analyzed the interaction of ULK1 with HSC70, which is an adapter for CMA. GST-ULK1 or GST vector was cotransfected with HA-HSC70 into HEK293T cells. A pull-down assay was performed and analyzed by WB using an anti-HA antibody. ULK1 bound HSC70 (Fig. 6d).

When chaperone-mediated autophagy is enhanced, macroautophagy is weakened. Upon knockout of Atg5, chaperone-mediated autophagy increased[18]. When ULK1 was knocked out, the amount of LAMP2a and HSC70 increased, which strongly indicated that CMA was enhanced (Supplementary Figure 8d, e). Our results are similar to what was found when knocking out Atg5[18]. Therefore, CMA may adjust the overall autophagy level of the cell and balance macroautophagy[18,19].

The HSC70 associated CMA pathway uses the KFERQ-like motif. The KFERQ motif usually contains five amino acids, including a key glutamine (Q) residue that is preceded by four other residues consisting of an acidic (D or E), a basic (K or R), and a hydrophobic residue (F, I, L, or V)[20]. We found two KFERQ-like motifs in ULK1 and mutated these motifs for an IP assay. The results showed that 227-QDLRL-231 is responsible for their binding (Fig. 6e). To explore whether ULK1 phosphorylation affects the interaction, we transfected ULK1 variants with Lamp2a or HA-HSC70 and performed an IP assay. We observed that ULK1 S423D bound more Lamp2a and HSC70 in cells (Fig. 6f, g). HeLa cells were treated with DMSO, Phorbol 12-myristate 13-acetate (PMA, a PKC activator) and GÖ6983, and the cell lysates were used for an IP assay. ULK1 bound more HSC70 and Lamp2 in PMA treatment cells; however, GÖ6983 decreased their interaction (Fig. 6h). These data demonstrated that ULK1 phosphorylation increased the interaction of ULK1 with HSC70 and Lamp2. PMA treatment triggers faster ULK1 degradation in WT cells than in *LAMP2-KO* cells (Fig. 6i). We also detected ULK1 in *Lamp2a* KD cells. PKM2, which is degraded through CMA, was used as a positive control[20]. ULK1 accumulated in both *LAMP2-KO* and *LAMP2a-KD* cells (Supplementary Figure 9a). Colocalization of ULK1 with LAMP2 or LAMP2a decreased under serum starvation (Supplementary Figure 9b, c), suggesting that ULK1 is degraded through the CMA pathway. ULK1 Q227D228AA accumulated in MG132 and PMA treatment but ULK1 WT did not, suggesting that ULK1 Q227D228AA could not be degraded by the lysosome. The triple mutant S423DQ227D228AA is much more stable than S423D (Fig. 6j). Quantification of the remaining ULK1 is shown in Fig. 6k. Combined with previous reports[18,19], CMA is able to not only compensate for the reduction of macroautophagy but also act as a counterbalance in macroautophagy through PKCα-mediated ULK1 degradation.

## Discussion

ULK1 plays a key role in the initiation of autophagy through the ULK1/Fip200/Atg13 complex[21]. ULK1 also regulates autophagy via phosphorylation of the BECN1/VPS34/VPS15 complex[8]. However, the specific role of ULK1 in the late stage of autophagy is unclear. We demonstrated that ULK1 regulates autophagosome–lysosome fusion mediated by STX17, which is regulated by PKCα phosphorylation. Unphosphorylated ULK1

---

**Fig. 4** ULK1 physically interacts with STX17. **a**, **b**, **c** Interaction between ULK1 and STX17 were determined using overexpression (**a**) and endogenous (**b**, **c**) proteins to perform IP assay followed by WB with indicated antibodies. **d** GST-ULK1 was cotransfected with GFP-STX17 fragments into HEK293T cells for 24 h. Pull down with glutathione beads was performed and analyzed with GFP antibody. **e** The GST-ULK1 truncations were transfected into HeLa cells for 24 h, and pull down with glutathione beads and analyzed with STX17 antibody. **f** GST-ULK1 279–525 and His-STX17 157–275 were purified from bacteria and incubated with glutathione beads for 4 h. After washing, the elution was analyzed with His antibody. **g** Myc-ULK1 or Myc vector was cotransfected with GFP-STX17 for 24 h. PLA assay was performed, and the signals (red) were detected under confocal microscopy. Scale bar, 10 μm. **h** The mice were injected with GÖ6983 (22.0 μg per mouse) or PBS for 7 days via tail vein. IP was performed of liver lysates with STX17 antibody. The IPs were detected by ULK1 antibody. The indicated antibodies were used to confirm drug efficiency. **i** HeLa cells were transfected with GFP-STX17 and Myc-ULK1 for 24 h with/out serum starvation (S.S.). PLA assay was performed with Myc and GFP antibodies. The fluorescence signal (red) was observed under confocal microscopy. Scale bar, 10 μm. An average signal was quantified with a minimum of 20 cells and is shown in the right bottom panel. **j** The representative images of PLA assay of GFP-STX17 cotransfected with Myc-ULK1 WT, S423A, S423D, or Myc vector into HeLa cells for 24 h. Scale bar, 10 μm. An average signal was quantified with a minimum of 20 cells and is shown in the left top panel. All values are means ± SEM of three independent experiments. Student's *t*-test (unpaired); *$p < 0.05$, **$p < 0.01$, ***$p < 0.001$. WCL: whole-cell lysate

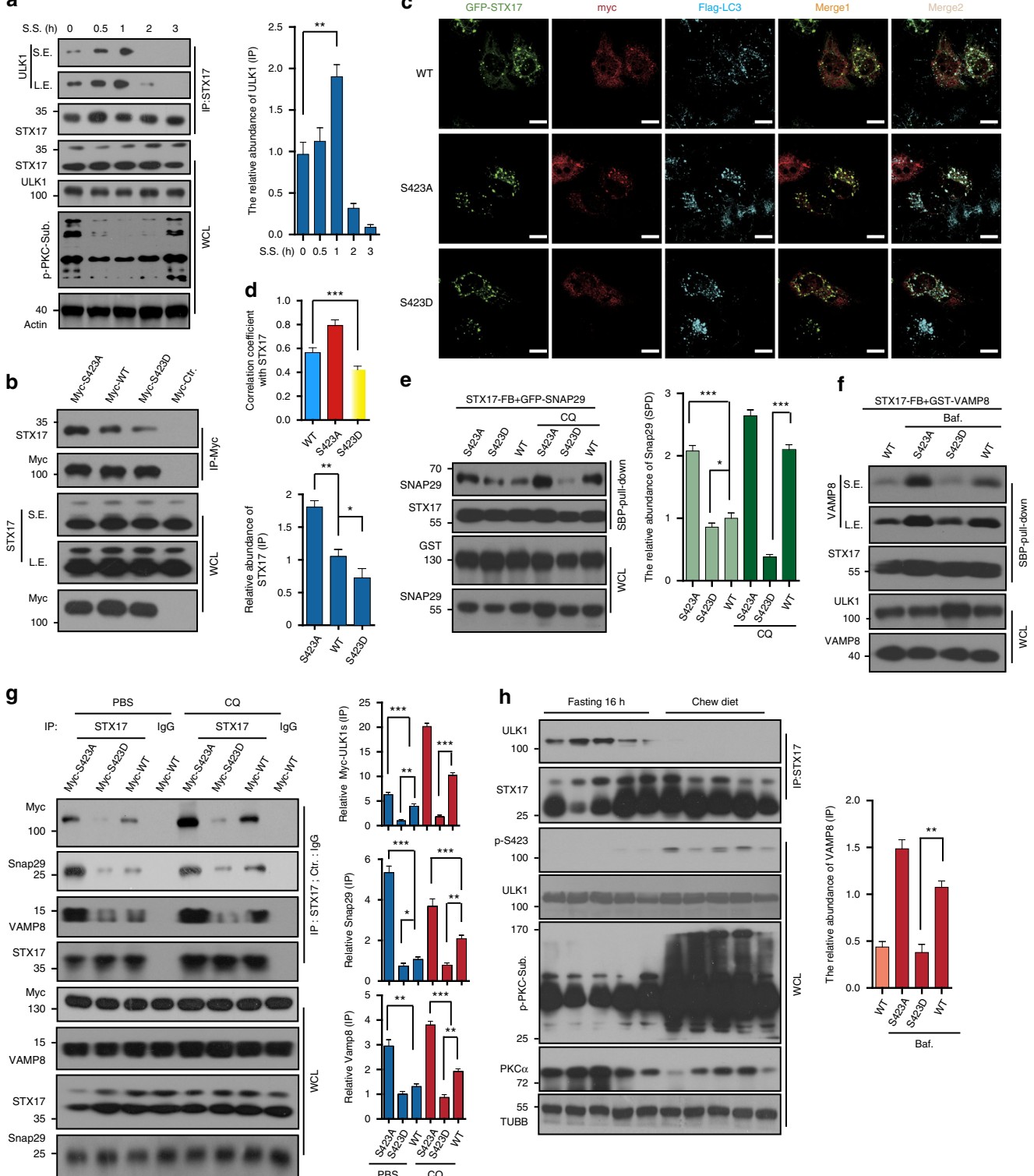

**Fig. 5** Phosphorylation of ULK1 influences the STX17 complex. **a** HeLa cells were starved and the lysates were performed with IP assay via STX17 antibody and analyzed with ULK1 antibody showed in the right top panel. IPed ULK1 is quantified and showed in the right bottom panel. **b** ULK1 S423D decreased binding to STX17 compared to WT and S423A by IP assay. In addition, IPed STX17 was quantified. **c, d** GFP-STX17, Flag-LC3, and Myc-ULK1 variants were cotransfected and detected by immunostaining in HeLa cells ($n = 20$), and the images were quantified (**d**). Scale bars, 10 μm. **e, f** ULK1 S423D affects interactions between STX17 and SNAP29 or VAMP8 with cotransfection and IP assay. The results were analyzed with the indicated antibodies. Bound SNAP29 and VAMP8 were quantified. **g** myc-ULK1 variant transfected HeLa cells were treated with CQ (100 nM) or PBS. IP assay was performed with STX17 antibody or IgG and detected by indicated antibodies. IPed proteins were quantified. **h** Lysates from the livers of fasting or normal chew mice were performed by IP assay with STX17 antibody and analyzed with ULK1 antibody. All values are means ± SEM of three independent experiments. Student's $t$-test (unpaired); *$p < 0.05$, **$p < 0.01$, ***$p < 0.001$. Baf: bafilomycin a1, WCL: whole-cell lysate

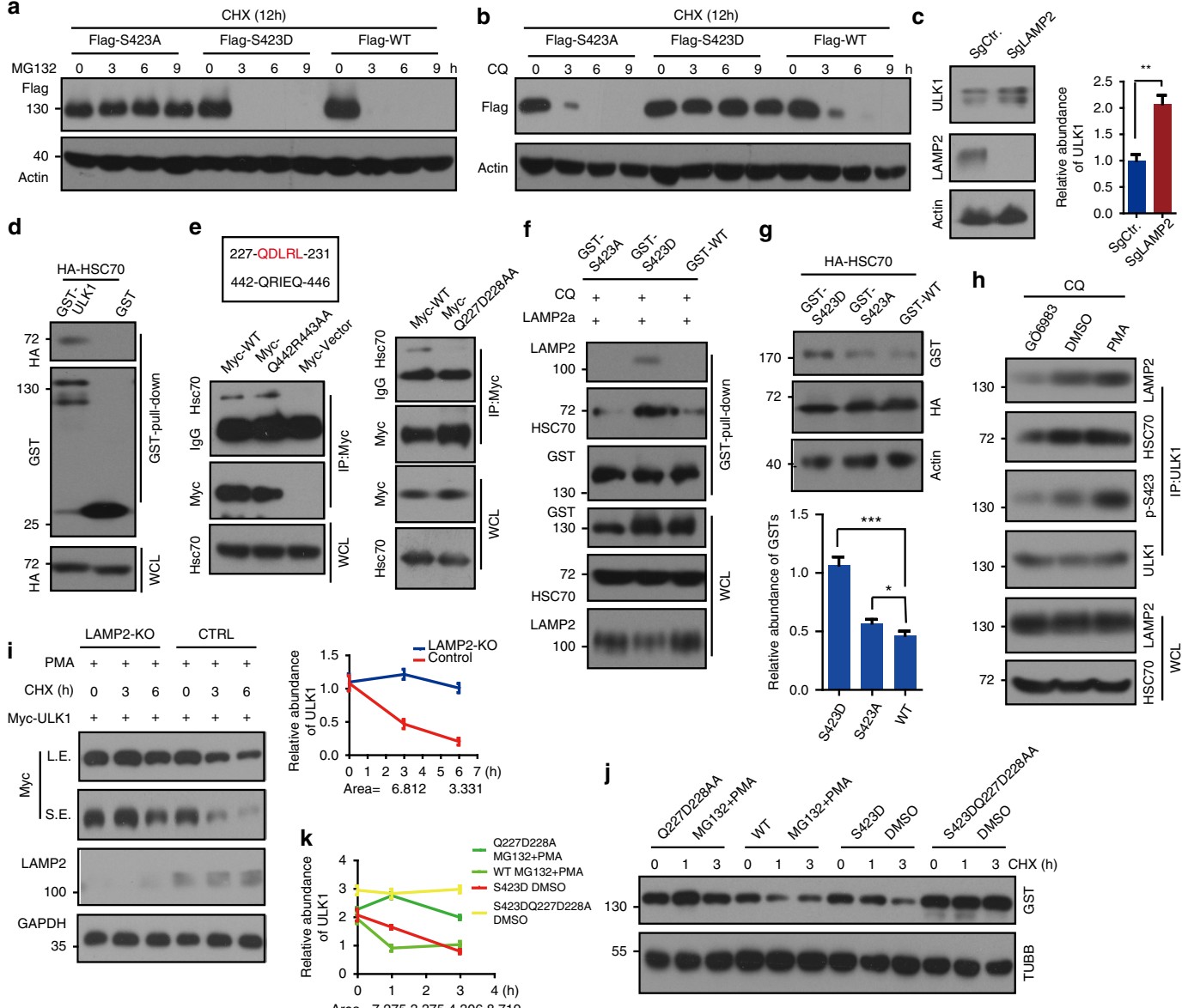

**Fig. 6** Phosphorylation of ULK1 is degraded through chaperone-mediated autophagy. **a** MG132 (proteasome inhibitor, 20 μm) prevents ULK1 S423A degradation but not degradation of ULK1 WT or S423D. Stable ULK1 WT and mutant HeLa cells were pretreated with cycloheximide (CHX, 20 μg mL$^{-1}$), an inhibitor of protein synthesis, for 3 h. Then, the cells were treated with MG132 for a time course in the presence of CHX. The remaining ULK1 was detected by WB. **b** CQ inhibits ULK1 S423D degradation. The procedure is similar to Fig. 6a, but MG132 is replaced by CQ. **c** Accumulation of more ULK1 in *Lamp2-KO* cells. The abundance of ULK1 was quantified. **d** The interaction of ULK1 and HSC70 was determined by pull-down assay with overexpression of GST-ULK1 and HA-HSC70. **e** The HSC70 binding motif of ULK1 is between Q227 to L231, and the result was verified by double mutations and semi-endogenous co-IP assay by HSC70 antibody. **f** ULK1 S423D binds more HSC70 and LAMP2a than its counter partners. **g** Overexpression of HSC70 promotes ULK1 WT and S423Ds degradation, not S423A degradation. **h** PKCα activity increased the affinity of ULK1 to HSC70 and LAMP2. The results were detected with the indicated antibodies. WCL whole-cell lysate. **i** ULK1 degraded slowly in *LAMP2-KO* cells with PMA (a PKC activator) treatment. Myc-tagged ULK1 was transfected into the above two cell lines with the indicated drugs and time course. The remaining ULK1 was quantified. **j** Mutant ULK1 Q227D228AA prevents p-ULK1 degradation through the lysosome. **k** The quantification of Fig. 6j. All values are means ± SEM of three independent experiments: *$p < 0.05$, **$p < 0.01$, ***$p < 0.001$. WCL: whole-cell lysate, L.E.: long exposure, S.E.: short exposure

enhanced fusion of the autophagosome to the lysosome by increasing the interaction with STX17 and promoting STX17/ SNAP29/VAMP8 formation. We also found that phosphorylated ULK1 is degraded through the CMA pathway to avoid participating in the fusion step (Fig. 7).

Autophagy is a continuous process, including initiation, expansion, autophagosome formation, and fusion to the lysosome. ULK1 plays a central role during the initiation stage of autophagy. The ULK1/FIP200/ATG13/ATG101 complex is a primary structure for autophagy that responds to nutrient conditions. When nutrient conditions change, ULK1 is phosphorylated by mTORC1 or AMPK to regulate autophagy. As a serine/threonine kinase, ULK1 also phosphorylates its substrate to regulate autophagy. These substrates include BECN1, VSP34, ATG13, and ATG101[5]. All reported functions of ULK1 are associated with the autophagy initiation stage, but none have reported its role in fusion of the autophagosome to the lysosome.

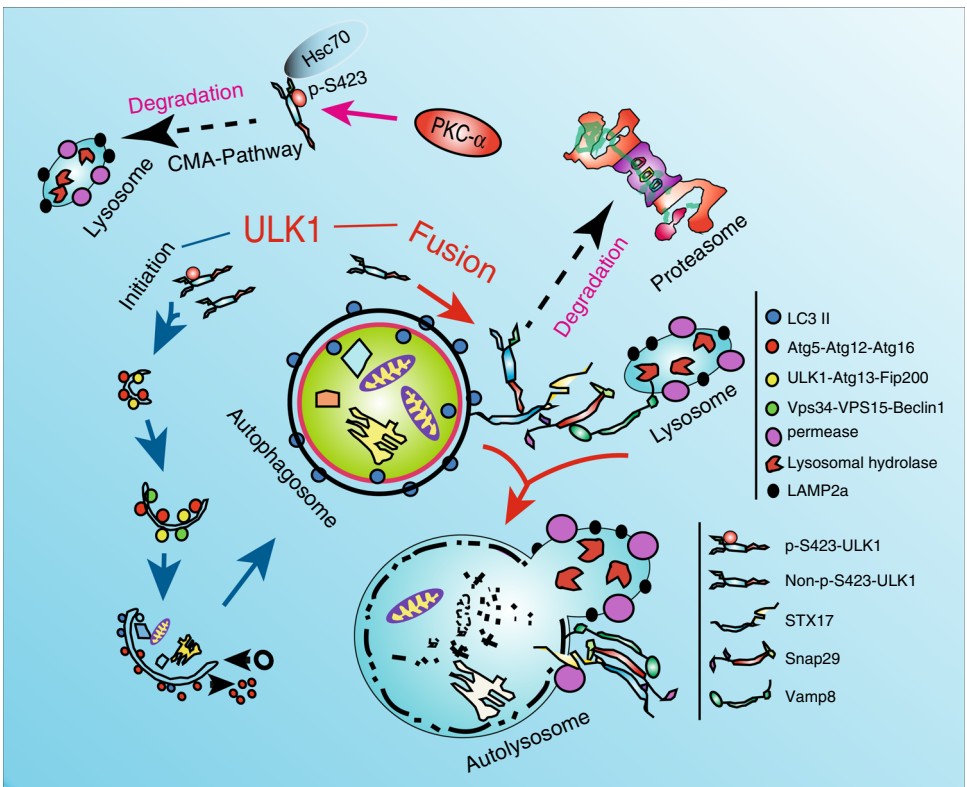

**Fig. 7** Model of ULK1 regulating fusion of macroautophagy and linking chaperone-mediated autophagy to macroautophagy. Unphosphorylated ULK1 recruited STX17 and increased STX17's affinity towards SNAP29. PKCα phosphorylation of ULK1 attenuates autophagosome–lysosome fusion by reducing the affinity of ULK1 for STX17. Phosphorylation of ULK1 enhances its interaction with HSC70 and increases its degradation via CMA

Mutual fusion between membrane vesicles is mediated by the tSNARE and vSNARE gene families[22,23]. The fusion of autophagosomes to lysosomes occurs through tSNARE STX17 on autophagosomes, which forms a membrane complex with vSNARE VAMP8 on lysosomes via SNAP29[8]. ATG14L regulates autophagy flux by interacting with BCN1/VPS34/VPS15 and mediates autophagosome–lysosome fusion through STX17[9,10]. We also demonstrated that another protein, ULK1, plays dual roles in autophagy flux and fusion, indicating that protein factors, which are involved in the early stage of autophagy, also play a role in the late stages of autophagy. In the future, the functions of factors in different stages should be analyzed. Here, we found that ULK1 acts a key regulator in the fusion step by manipulating phosphorylation. We tried to deeply explore the mechanism of how non-phosphorylated ULK1 regulates the interaction of STX17 and SNAP29. We tried to determine whether ULK1 can phosphorylate STX17 by in vitro kinase assay followed by phos-tag gel examination. We did not observe the phosphorylation of STX17. ULK1 exerts its effect on autolysosome formation through transient protein-protein interactions between non-S423-ULK1 and STX17. The transient binding of unphosphorylated ULK1 to STX17 promoted the interaction of STX17 and Snap29. ULK1 does not form a complex with STX17 and Snap29. After STX17 interacts with Snap29, ULK1 is dissociated from STX17 and is degraded through the proteasome pathway (Fig. 7).

On the regulation of macroautophagy in PKC signaling pathway,Recent studies showed conflicting findings. PKC activation is required for palmitic acid induced autophagy[24]. PKC inhibitors induce autophagy[7]. Both PKCα and PKCβ can suppress autophagy process[24,25].

Autophagy inhibition by different active kinases (mTOR and PKC) in nutrient-rich conditions is achieved by simultaneous changes at different stages of autophagy. Under nutrient-rich conditions, both mTOR and PKC are active, and autophagy is inhibited. mTOR inhibits the early stage of autophagy; PKC inhibits the late stages of autophagy not only through phosphorylation of LC3[7], but also phosphorylate ULK1 in our study (Supplementary Figure 10).

With sufficient nutrients, ULK1 is phosphorylated and degraded by the CMA pathway to reduce macroautophagy. CMA and macroautophagy compensate for each other, but how they are linked together is unknown[18,19]. Our data proved that ULK1 is the key regulator linking these pathways. Active PKCα represents high nutrient conditions for cells. Consequently, the cells require only a very low level of autophagy. CMA can reduce the autophagy level by degrading ULK1, which is phosphorylated by PKCα. In summary, our findings highlight the dual roles of ULK1, which acts both as a protein kinase and as a target for phosphorylation during multiple phases of autophagy, including the initiation of autolysosome formation of macroautophagy and chaperone-mediated autophagy. These results greatly affect our understanding of dual functioning proteins in the control of macroautophagy. Demonstration of the mutual regulation of macroautophagy and CMA integrated by the protein kinase activity of PKCα revealed that PKCα has a significant role in maintaining the balance of autophagy and is an attractive molecular target for autophagy-related human diseases.

## Methods

**Animals**. All animals were maintained and used in accordance with the guidelines of the Institutional Animal Care and Use Committee of the Institute for Nutritional Sciences. All of the experimental procedures were carried out in accordance with the CAS ethics commission with approval number 2015-AN-6.

**Antibodies and reagents**. Phospho-AMPKα T172 (2531; 1:1000), AMPKα1 (2532; 1:1000), phospho-ULK1 S757 (14202; 1:1000), phospho-beclin 1 S15 (13825; 1:1000), beclin 1 (3495; 1:1000), PKCα (2056; 1:1000), ULK1 (8054; 1:1000),

ubiquitin (3933; 1:1000), 4E-BP1 (9452; 1:1000), phospho-4E-BP1 Thr37/46 (2855; 1:1000), mTOR (2983; 1:1000), phospho-mTOR S2448 (5536; 1:1000), phospho-(Ser/Thr) PKC substrate (6967; 1:1000), β-tubulin (2146; 1:3000), HA (2999; 1:3000), GFP (2956; 1:2000), and Myc (2278; 1:2000) antibodies were from Cell Signaling Technology, Danvers, MA. LC3 antibody (NB100-2220; 1:7000) was from NOVUS Biologicals, Littleton, CO. p-Atg13 S318 (600-401-C49S; 1:1000) was from Rockland Immunochemicals, Limerick, PA. β-Actin antibodies (sc-47778; 1:10000) were from Santa Cruz Biotechnology, Dallas, Texas. P62 (SQSTM1) antibody (ab56416; 1:4000), SNAP29 (ab56566; 1:1000), LAMP2a (ab18528; 1:3000), and VAMP8 (EP2629Y; 1:1000) were from Abcam, Cambridge, MA. GST (E022040-01; 1:10000) was from EARTHOX Life Sciences, Millbrae, CA, and Flag (SLBJ7864V; 1:2000) and STX17 (HPA001204; 1:1000) were purchased from SIGMA-Aldrich, St. Louis, MO. FIP200 (17250-1-AP; 1:1000) was from Proteintech, Rosemont, IL. Total p-S, T (612549; 1:4000) antibody was from BD Biosciences, San Jose, CA. Anti-phospho ULK1 S423 antibody (1:100) was generated and affinity purified by ABclonal Technology, Wuhan, China.

**DNA constructions.** The genes with HA, GST, Myc, Flag, and SBP tags were cloned into pRK5 via Sal I and Not I restriction enzyme from cDNA. The lentivirus of doxycycline-inducible and Flag-tagged ULK1 was constructed via PCR and infusion methods. GST-tagged ULK1(279-525AA) was inserted into the pGEX-4t-2 vector via the PCR method, and STX17(157–275) with a C-terminal 6*His was inserted into pET-26b( + ) via infusion technology (VAZAME, C112-01/02). The shRNA sequences (Table 1) were cloned into the pGIPZ vector via inverse PCR.

The sgRNA sequences (Table 2) were cloned into the CRISPR/cas9 system via PCR.

**Cell culture and transfection.** HEK293T, HEK293, HEK293FT, 293A, and HeLa cells were purchased from Cell Bank (Shanghai Institutes for Biological Sciences, CAS, China). B16F10 and U87 cells were purchased from ATCC (American Type Culture Collection). These cell lines were cultured in DMEM (HyClone, SH30243.01B) medium with 50 μg mL$^{-1}$ penicillin/streptomycin (Gibco, 15140–122) and 10% FBS (HyClone, SV30084.03). HEK293 and HeLa stable cell lines were cultured in media with additional puromycin (SIGMA, 58-60-6) or hygromycin (CALBIOVHEM, 400050). The serum starvation experiments performed in this manuscript were switched from normal media to media without FBS. For transient transfection, recombinant DNA plasmids were transfected into related cells using polyethyleneimine (PEI, SIGMA). Approximately 1 μg of plasmid and 1.5 μL of PEI (1 mg mL$^{-1}$, pH 7.0) were separately added into 250 μL of serum-free DMEM for 5 min, then mixed together for 15 min and put into the cell dish. After 6 h, the media was changed. Cells were used for WB or immunostaining at indicated times.

**Table 1 Sequences of shRNAs**

| | |
|---|---|
| shCTR: | GCTTCTAACACCGGAGGTCTT |
| shPKCα1: | CGACGACTGTCTGTAGAAA |
| shPKCα2: | GGATTGTTCTTTCTTCATA |
| shPKCα3: | CACATCCAGGCAAGAACTA |
| shULK2-1: | TTCTACAGCACAATTATAG |
| shULK2-2: | TTCGCTTGCAAATAATCTG |
| shULK2-3: | TTCATGTACAACACCAGCT |
| shSNAP29-1: | GACAAGTTAGATGTCAACATAA |
| shSNAP29-2: | ATCATGCAGAAGCATCAATTAA |
| shSNAP29-3: | AAAGAAGCTATAAGTACAAGTA |
| shVAMP8-1: | ATCGCAGAAGGTGGCTCGAAAA |
| shVAMP8-2: | GCTCGAAAATTCTGGTGGAAGA |
| shVAMP8-3: | GTGGAGGGAGTTAAGAATATTA |
| shULK2-1: | TTCTACAGCACAATTATAG |
| shULK2-2: | TTCGCTTGCAAATAATCTG |
| shULK2-3: | TTCATGTACAACACCAGCT |
| shLAMP2a: | TGCAGTGCAGATGACGACA |

**Table 2 Sequences of sgRNAs**

| | |
|---|---|
| sgLAMP2: | ATAGCAGTGCAGTTCGGACC |
| sg1ULK1: | GGAGAACTCGAACTTGCCCA |
| sg2ULK1: | TCGCTGACTTCGGCTTCGCG |

**ULK1 KO and ULK2 KD Cells.** After HeLa cells were infected with *ULK1*-SgRNA1 lentivirus and selected with puromycin, we obtained seven monoclonal knockout cells and numbered them 1–7. We designed three shRNA sequences and used pGIPZ-sh2-*ULK2* lentivirus to infect the *ULK1-KO* cell line and screened the cells by hygromycin. For the transfection experiments related to ULK1, we used the mutation method on the cutting site of the ULK1s plasmids to avoid CRISPR/Cas9′s influence.

**Lentiviral preparation and viral infection.** Lentiviral shRNAs of pGIPZ was constructed via an inverse PCR method[26]. The pGIPZ shRNA vectors encoding shRNAs were cotransfected with lentiviral packaging vectors pSPAX2 and pMD2.G into HEK293FT cells. The pCDH overexpression vectors were cotransfected with lentiviral packaging vectors pMDL, VSVG, and Rev into HEK293FT cells[27]. Viruses were collected after 48 h transfection. And U87, HeLa or HEK293 cells were infected with the collected viruses for 12 h. Some cells were selected as stable cell lines following manufactory's manual. The lentivirus vectors that contained doxycycline inducing components and Flag-tags were constructed based on the backbone of pGIPZ. Consequently, lentiviral preparation and viral infection were done in the same manner as the pGIPZ lentivirus system[28].

**Protein purification.** Briefly, the bacterial expression system for His-tagged STX17 (157–275AA) and GST-recombinant ULK1 (279–525AA) were transformed into BL21 codon plus (Stratagene). Protein induction was under the condition of 0.5 mM IPTG at 16 °C. Bacterial cells were resuspended in cold PBS containing 5 mM 2-mercaptoethanol, 1% Triton X-100, 1 mM PMSF, and 2 mM EDTA followed by ultrasonication. Then, the GST-recombinant proteins and His-tagged proteins were purified following the manufactory protocol. His-tagged PKCα was commercial available (PROSPEC, Catalog Number: PKA-218).

**Immunoprecipitation and western blot.** The cells or tissue were homogenized in IP/WB Lysis buffer (20 mM Tris-HCl, 150 mM NaCl, 1 mM EDTA, 0.5 mM DTT, 0.2% NP-40, 0.5% Triton X-100, 10% Glycerol, pH 7.4) contained phosphatase inhibitor cocktails 1/2 (Roche, 04906837001) and protease inhibitor cocktail (Roche, 04693132001). Whole-cell lysate was centrifuged for 10 min at 4° and the supernatant used for immunoprecipitation via Glutathione Agarose (Thermo Fisher Scientific, 16102), Streptavidin Beads (genscript, L00353) or anti-Flag M2 Affinity Gel (Sigma, A2220), followed by incubation at 4 °C overnight. The related beads were washed six times with IP buffer and boiled in SDS-PAGE buffer for 5 min at 95 °C. The samples were separated by SDS-PAGE Gel and transferred onto PVDF membranes (Bio-Rad). Western blot experiments were performed via the indicated antibodies and visualized by Super-Signal West Pico Chemiluminescent substrate (Pierce Chemical).

**Immunofluorescence staining.** HeLa cells were cultured in 4-well chamber slides and transfected with the indicated plasmids via PEI for 24 h. The cells were then fixed in 4% paraformaldehyde for 20 min at room temperature, then permeabilized by PBS with 0.1% Triton X-100 for 15 min at room temperature. The HeLa cells were washed four times with PBS, then blocked with 10% BSA in PBS for 1.5 h at RT. Then, cells were incubated with anti-LC3 (1:500), LAMP2(1:200), LAMP2a (1:200), Myc (1:100), GFP (1:500), and Flag (1:500) antibodies for 1.5 h at 25 °C. After washing two times with PBS, the cells were incubated with Alexa Fluor 568-conjugated anti-mouse IgG, Alexa Fluor 647-conjugated anti-mouse IgG or Alexa Fluor 488-conjugated anti-rabbit IgG antibody (Invitrogen, 1:1000) for 1.5 h. Images were obtained by DPS-PSW software with an Olympus microscope (Olympus; IX71). Images were photographed at random positions for each condition.

**In vitro PKC kinase assay.** The PKC kinase buffer contains 50 mM HEPES, 0.01% Tween 20, 10 mM MnCl2, 1 mM EGTA, 2.5 mM DTT and 0.1 mM ATP, pH 7.4. His-tagged PKCα was purified from Sf9 insect cells (PROSPEC, Catalog Number: PKA-218). Approximately 0.5 μg of purified His-PKCα and 2 μg of GST-tagged recombinant substrate were added to the reaction buffer for 60 min at 30 °C. All reactions were stopped by adding 5 × SDS loading buffer and boiled for 10 min at 95 °C for WB analysis[21].

**Transmission electron microscope.** After 3 h treatment with CQ or DMSO, WT, S423A, and S423D HeLa cells were harvested by trypsin digestion and fixed with 2.5% glutaraldehyde for 2 h at 4 °C, followed by treatment with 1% osmium tetroxide in 0.1 M cacodylate buffer for 1 h. The fixed HeLa cells were dehydrated with sequential gradient washes from 30 to 100% ethanol and then immersed in epoxy resin afterwards. Ultrathin sections were placed on carbon-coated copper grids and counterstained with uranyl acetate and lead citrate. Images were taken with an FEI Tecnai G2 Spirit transmission electron microscope.

**qPCR.** The samples were obtained using RNAiso Plus (TAKARA) following the company's protocol. cDNA was synthesized using the PrimeScript RT reagent kit with gDNA eraser (TAKARA). mRNA levels were determined using SYBR Premix ExTaq (TliRNaseH plus) (TAKARA) and an ABIPRISM 7900HT Sequence detector (Perkin Elmer). The detailed process is 95 °C for 30 s and 40 cycles of

95 °C for 5 s, followed by 60 °C for 30 s. The qPCR results were normalized against GAPDH as a control.

**LC-MS/MS.** GST-recombinant ULK1 (279-525AA) was purified from BL21 codon plus. After in vitro PKC kinase assay, the reaction was stopped by adding 5 × SDS loading buffer and boiled for 10 min at 95 °C before SDS-PAGE with Coomassie blue staining. The band of GST-recombinant ULK1 was cut as the LC-MS/MS sample. The gel was solubilized and reduced in a buffer containing 7 M urea, 2 M thiourea, 10 mM HEPES, 1 mM sodium orthovanadate, 5 mM sodium fluoride, 5 mM β-gly-cerophosphate, and 10 mM DTT, pH 8.0, then alkylated with 15 mM iodoacetamide. The samples were diluted to a final urea concentration of 2 M, digested with trypsin (Promega, USA) at 37 °C (1:100 w/w) overnight, then desalted by reversed-phase C18 Sep-Pak cartridge (Millipore, USA). The samples were analyzed on an EASY-nLC1000 LC system (Thermo Scientific) coupled to the Q-Exactive mass spectro-meter (Thermo Scientific). Peptides were loaded on an in-house packed C18-column (15 cm, 75 μm I.D., and 2-μm particle size) with a gradient of 12–32% buffer B (98% acetonitrile and 0.1% acetic acid) for 45 min followed by a 5-min wash with 95% buffer B. Full scan MS spectra were acquired at a resolution of 70,000 with an AGC target value of $3 \times 10^6$ and a maximum injection time of 100 ms. MS/MS spectra were acquired at a resolution of 17,500 with a target value of $1 \times 10^5$ and a maximum injection time of 50 ms. The raw files were processed by MaxQuant software (version 1.5.3.8) with searches against the Uniprot human database and a variable mod-ification of serine, threonine and tyrosine phosphorylation.

**PLA technology.** Adherent HeLa cells on the glass slides were transfected with the indicated plasmids. After fixation, cells were pretreated and blocked with 5% BSA. The requirements of the primary antibodies were used for the slides. The secondary antibodies, which were conjugated with oligonucleotides (PLA probe MINUS, SIGMA DUO92004 and PLA probe PLUS, SIMGA DUO92002), were added to the glass slides for 1 h at 37 °C. After incubation, a system consisting of two oligo-nucleotides and a ligase were added to the system for 30 min at 37 °C. If the oligonucleotide probes were in close proximity, they formed a closed circle. Then, the slides were incubated for amplification by the PLA probe (SIGMA DUO92008) and the polymerase for 100 min at 37 °C. The signal was seen as a unique fluor-escent spot under confocal microscopy.

**In vitro fusion experiment.** HeLa cells ($1 \times 10^6$) were transfected for 24 h with mRFP–LC3 or stained with lysosome tracker (LysoTracker Green DND-26-Special Packaging) for 3 min. HeLa cells were scraped into 600 μL of HB buffer (homo-genization buffer: 250 mM sucrose and 3 mM imidazole, 20 mM HEPES (pH 7.4) and protease inhibitors) and were lysed by 50 passages through a 27 G needle. After centrifugation at 900×g and 4 °C for 5 min to pellet nuclei and cell debris, the supernatant was collected, and 10 μL was mixed in duplicate for 60 min in the presence of an ATP regenerative system (10×: DHM buffer: 625 mM HEPES, 75 mM magnesium acetate and 10 mM DTT (pH 7.4, with KOH), 1 M potassium acetate, 100 mM ATP, 800 mM creatine phosphate, 4 mg mL$^{-1}$ creatine kinase (=3200 U mL$^{-1}$), 250 mM glucose and 100 mM ATP) at 37 °C in low adhesive Eppendorf tubes in a total volume of 40 μL with 2 μg of recombinant GST-ULK1s (from 293Ft cells). After the reaction, samples were fixed with 2% paraformalde-hyde for 20 min (1:1 with 4% PFA), centrifuged to remove the fixative (13,200 rpm for 7 min), resuspended in distilled water, and mounted on glass coverslips with neutral balsam for confocal observation[29].

**Statistical analysis.** Statistical significance was measured via unpaired and two-tailed Student's tests and is presented as follows: $*p < 0.05$, $**p < 0.01$, and $***p < 0.001$. All error bars indicate SEM.

**Data availability.** The authors declare that the data supporting the findings of this study are available within the article and its Supplementary Information Files. All other relevant data supporting the findings of this study are available on request.

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

## Acknowledgements

We thank Dr. Yi hua. Gu (Shanghai Jisheng Hospital) and Dr. Zachary Ende (Emory University) for English editing. This work was supported by grants from the Ministry of Science and Technology of China (973 Program; 2014CB910500) and the National Natural Science Foundation of China (81172231) to Z.L.

## Author contributions

C.W., H.W., D.Z. and W.L. performed the biological and biochemical experiments. Z.L. and C.W. conceived the project, designed the experiments, and wrote the manuscript.

R.L. and D.X. provided some experimental samples, and L.D. offered some advice on the project. L.L. analyzed the mass spectrometry data.

## Additional information

**Competing interests:** The authors declare no competing interests.

