## [Peer Review File · Nature Communications]

Reviewer #1 (Remarks to the Author):

The paper suggests a new and attractive role of ULK1 in autophagosome/lysosome fusion, promoting to fusion of both membranes by promoting the interaction of the Stx17-SNAP29-VAMP8. According to the authors, the phosphorylation state of ULK1 at S423 might regulate this step, unphosphorylated-ULK1 enhances Stx17 interaction with its partners and favors fusion, while phosphorylation of ULK1 at its S423 promotes its degradation by CMA.

The hypothesis is appealing, and a lot of work has been done, however this work presents and endless number of technical errors, uncompleted experiments, missing information and inaccuracies that overall cannot sustain the conclusions and make this paper not suitable for publication as it is.

GENERAL COMMENTS:

- Most of the work presented in this manuscript is based in overexpression of endogenous proteins fused to different tags (HA, myc, GFP, GST, Flag...) followed by immunoprecipitation or pulldown. However in almost all the experiments the appropriate controls are missing and consequently the results are not reliable. For any IP/pulldown, as a control, is necessary to include a sample without the target protein (empty plasmid, only with the tag) and in parallel show that the immunoprecipitated protein is present only in the lane where the interaction between the two proteins is positive, however this type of control is missing in almost all the figures presented, consequently is hard to be convinced that the interaction detected is a specific and real interaction.

-As mentioned, almost all work presented is based in protein-protein interaction, and always using an artificial overexpressed system. Endogenous conditions or alternative systems (PLA, colocalization) should be used to confirm that these interactions are physiologically relevant.

-The information given by authors regarding each experiment and including the main text, the figure legends, and material and methods is totally insufficient to understand each figure. There is a huge lack of details and information that make almost impossible to fully understand and identify all data presented. This is one of the main concerns of this manuscript.

-Almost all results are presented as a single image of a western blot. There is any quantification corresponding to different independent experiments to confirm that the changes observed are significant, especially when increase/decrease in the levels of proteins is the main result. Figure 3a is the only exception.

SPECIFIC COMMENTS:

-Fig 1a: authors say that to determine the phosphorylation by PKC, different autophagy-related proteins were overexpressed, however only results on ULK1 are presented. Did authors checked other proteins as they say or they directly checked ULK1? Why they say that "only the ULK1 showed a signal"?

- Fig1a is impossible to understand what are the different lanes of the WB presented, what means "WLC"? What are the different bands detected in each lane and each blot? What type of sample is each lane? The information is missing in the main text and in the legend. This type of error is repeated all along the manuscript.

-Fig1b: The p-PKC-substrate antibody is not specific to conclude that the kinase phosphorylating ULK1 is PKC-alpha, it can be any other PKC kinase.

-Why in almost all experiments in this section, ULK1 proteins are overexpressed and immunoprecipitated against the same ULK1? Why changes in phosphorylation are not showed directly under endogenous levels without overexpression?

-Fig S1c and others: control with empty Myc/Flag/HA-construct are always missing.

-Fig S1e and S1f are interchanged.

-Fig S1f: why only fragment 2 pulldowns HA-ULK1? If the interaction region is between 167 and 338, fragment 4 or fragment 5 should also pulldown Flag-ULK1.

-FigS1i: in main text, the sentence is cut off?

-Fig2: All negative controls in all IPs are missing.

-Fig2f: Staurosporine is a broad spectrum kinase inhibitor, is not specific for PKC-alpha, any other kinase (mTOR, AMPK...) might also be affected. Go6983 is a broad spectrum PKC inhibitor. Contrary to what authors say, P-Atg13 and P-Beclin1 are not decreased with the inhibitors, but P-mTOR does as well as p-PKC-sub. Authors don't explain the detail of what are the 5 types of different inhibitors used. The results in this experiment cannot support the conclusions.

-Fig S2a/b: these results are inconclusive, cannot confirm that " PKC could regulate autophagy independent of mTOR" . The quality of fig S2a/ S2b is not good to say that the effect of both inhibitors is higher than with rapamycin alone.

-Fig3: If ULK1- S423D (phospho mimic) is sent to degradation by CMA, why is it increasing autophagosome accumulation? Since p62 is not accumulated, we cannot attribute this effect to the inhibition of fusion. Authors do not give a clear hypothesis to explain why ULK1-S423D increase the number of autophagosome and this observation is quite spectacular.

-Fig4 and 5: same deficiencies as in previous figures, controls are missing and information to understand the details of the experiments are lacking.

-Fig 6a: authors say that MG132 can inhibit degradation of ULK1 WT and S423A by UPS but not S423D. However Fig 6a shows that UPS-dependent degradation prevented by MG132 treatment occurs only with S423A, but not WT or S423D.

-CMA degradation is mediated by isoform LAMP-2A. Knock-out cells used to study CMA process should only delete this isoform but not LAMP-2B and LAMP2C that might have other important roles in lysosomal biology.

-Fig 6c: other typical CMA substrates should be showed here as a positive control to confirm that this slight increase in ULK1 levels when total LAMP-2 is missing, is due to a lack of ULK1 degradation by CMA.

Reviewer #2 (Remarks to the Author):

In the manuscript by Wang et al., the authors investigated the role of ULK1 for a later step of the autophagy signaling process, i.e. the fusion of autophagosomes with lysosomes. The authors describe that ULK1 can be phosphorylated by PKCa at Ser423. This phosphorylation reduces the affinity of ULK1 to STX17. In contrast, the authors suggest that unphosphorylated ULK1 recruits STX17 to the autophagosome membrane, where it has a higher affinity for SNAP29. Phosphorylated ULK1 binds to HSC70 and becomes degraded through chaperone-mediated autophagy. Collectively, this study identifies ULK1 as an important regulator of late steps of the autophagy process. Although this is a very interesting study, I think major revisions are necessary in order to make this work acceptable for publication in NATURE COMMUNICATIONS. I think the authors need to address the following aspects.

Major points:

- 1) I think the authors need to better explain how PKC-dependent phosphorylation of ULK1 at Ser423 is regulated during autophagy. It appears that under nutrient-rich conditions, this site is phosphorylated (Figure 1I).
 - a. Does this in turn mean that ULK1 becomes constitutively degraded by CMA? According to figures 6A and 6B, ULK1 degradation is blocked neither by MG132 nor by chloroquine.
 - b. Does autophagy induction lead to PKC inactivation? Or does autophagy activate a specific phosphatase dephosphorylating this site?
 - c. I think the usage of staurosporine as PKC inhibitor should be avoided. This compound inhibits several members of the human kinome.
 - d. Apparently, ULK1 Ser423 phosphorylation is oscillating (Figure 1I). Does this mean that PKC becomes reactivated after 1.5 h of serum starvation?
- 2) To this reviewer, it is not clear how non-phosphorylated ULK1 regulates the interaction between

STX17 and SNAP29. In figure 5G, it appears that SNAP29 does not affect the interaction between ULK1 S423A and STX17. However, the interaction between wt ULK1 and STX17 can be reduced by SNAP29 overexpression. How can the authors explain this?

3) The authors only use serum starvation in figures 1I and 5A, and in figure 5I livers of fasting mice were used. With regard to the analysis of colocalization between ULK1 and STX17 on the one hand and between ULK1 and LAMP2 on the other hand by immunofluorescence, it would be interesting to perform these analyses under serum starvation.

Minor points:

1) The manuscript needs considerable proofreading. At present, the usage of the English language is not appropriate. For example line 242: "Partial of STX17 colocalized with ATG16 except STX17." (but there are several more passages)

2) Sometimes, the statements in the manuscript text do not reflect what can be seen in the figures:

a. Lines 160-162: The authors describe that phosphorylation of ATG13 and Beclin 1 is decreased after treatment with PKC inhibitors. In figure 2F, the signals appear to be increased. Furthermore, this indicates that ULK1 phosphorylation at Ser423 does affect ULK1 kinase activity, although the authors state that this is not the case.

b. Lines 188-190: the calculations of the ratios are not reflected by the diagrams shown in figure 3A

c. Line 282: "MG132 could prevent WT and S423A degradation". In figure 6A, WT degradation is clearly not prevented.

3) In some experiments, the authors make use of ULK1/2 KO HeLa cells. However, the authors do mention the origin of this cell line. I assume they were generated by CRISPR/Cas9, but in the Materials & Methods section only sgULK1 is mentioned. The authors should clearly indicate how this cell line was generated. Furthermore, the usage of this cell line should be indicated in the individual figures.

4) I do not request the quantification of all western blots. However, I think in some cases quantification is mandatory (e.g. figures 1I, 2D-F, supplemental figures S1C, S2, etc.)

5) Figure 1A: The different lanes need to be labeled.

6) Figure 1E: "1st" and "2nd" need to be explained in the legend.

7) Figure 3C: The red stars are almost not visible.

8) Figure 4: The authors identified aa 157-275 of STX17 and aa 279-525 of ULK1 responsible for the interaction. Can the authors comment why aa 1-216 of STX17 and aa 17-465 of ULK1 were used for the in vitro pull down experiment shown in figure 4F?

9) Supplemental Figure S1: From panel S1C, different binding affinities between WT/S423A/S423D and PKC are not evident (see minor point 4).

10) Supplemental Figure S1: I think labeling of "e" and "f" has to be switched.

11) Supplemental Figure S2: Here, the authors did not analyze autophagic flux (+/- BafA1 or CQ). Furthermore, quantification of this figure is essential (see minor point 4), since the difference between PKC/mTOR inhibition vs. mTOR inhibition alone is not obvious.

12) Supplemental Figure S4: The authors just present a workflow and a silver staining. There are

no indications how mass spectrometry was performed.

13) Supplemental Figure S6: The authors do not comment on the reduced co-localization between ULK1 S423A and LAMP2, although this does support their conclusion.

Reviewer #3 (Remarks to the Author):

There is considerable interest for the mechanisms controlling the biogenesis of autophagosomes. In this manuscript, authors report an interesting role of ULK1 protein in regulating SNARE-mediated membrane fusion for autophagosome maturation. This new role of ULK1 protein is modulated by the phosphorylation at S423 via PKC α . However, I would not recommend publication in Nature Communications.

First of all, ULK1 is a well-known marker for omegasome/phagophore before autophagosome formation. For example, a previous study suggested that ULK1 dissociates from the phagophore membrane together with other early ATG proteins [Autophagy 9, 1491, 2013]. How can ULK1 regulate SNARE-mediated autophagosome/lysosome fusion even though ULK1 is not on autophagosomes?

Secondly, this paper is not convincing. The major claims of this manuscript are not fully supported by direct evidences.

1. In Figure 5, since authors only tested the interaction between STX17 and SNAP29 or STX17 and VAMP8, it is hard for them to claim the influence on STX17-SNAP29-VAMP8 complex formation requiring three proteins simultaneously.
2. In the absence of standard in vitro fusion experiments involving proteoliposomes reconstituted with SNAREs, it is hard to conclude the impact on the membrane fusion step. A failure of recruiting STX17 to autophagosomes can be an alternative explanation for the accumulation of double-membrane structures.
3. All results to support "ULK1 can be phosphorylated by PKC α at Serine 423 in vivo and in vitro" were done by overexpression or Co-IP in the cell. There is no direct evidence to demonstrate the interaction between ULK1 and PKC α . As shown in Fig2f, the inhibition of PKC α can also come from reducing the mTORC activity, which should be excluded through appropriate control experiments.

Finally, this manuscript is not well prepared. Typos and fragments gave me a hard time to read. I was also frustrated by the low quality of images and poor arrangement of figure panels. For example, based on the poor resolution of Figure 3c, it's not easy for one to claim "accumulation of completed double membrane autophagosomes".

Reviewers' comments:

Reviewer #1 (Remarks to the Author):

The paper suggests a new and attractive role of ULK1 in autophagosome/lysosome fusion, promoting to fusion of both membranes by promoting the interaction of the Stx17-SNAP29-VAMP8. According to the authors, the phosphorylation state of ULK1 at S423 might regulate this step, unphosphorylated-ULK1 enhances Stx17 interaction with its partners and favors fusion, while phosphorylation of ULK1 at its S423 promotes its degradation by CMA. The hypothesis is appealing, and a lot of work has been done, however this work presents an endless number of technical errors, uncompleted experiments, missing information and inaccuracies that overall cannot sustain the conclusions and make this paper not suitable for publication as it is.

GENERAL COMMENTS:

- Most of the work presented in this manuscript is based in overexpression of endogenous proteins fused to different tags (HA, myc, GFP, GST, Flag...) followed by immunoprecipitation or pulldown. However in almost all the experiments the appropriate controls are missing and consequently the results are not reliable. For any IP/pulldown, as a control, is necessary to include a sample without the target protein (empty plasmid, only with the tag) and in parallel show that the immunoprecipitated protein is present only in the lane where the interaction between the two proteins is positive, however this type of control is missing in almost all the figures presented, consequently is hard to be convinced that the interaction detected is a specific and real interaction.

Response: Thanks for the great suggestion. For some previous figures, we did use empty tag-Vector as negative control but we did not describe clearly such as the labeled "Ctrl" represented "tag-vector" in **Fig. 4a** and **6d**.

According to your advice, we redo all of these IPs or pulldown assays including a control such as blank plasmid with tag or IgG. Besides, it is generally acceptable to show only the comparison of the IP experimental groups without a control group in many high-profile papers, such as *Nature*. 2011. PMID : 22056988; *Cell*. 2013. PMID : 24034250; *Immunity*. 2016. PMID: 27178468 . Based on your suggestion, we replace most of the figures to make them better. Thanks again for your understanding.

Figure 4

Figure 4. ULK1 physically interacts with STX17. (a) Interaction between ULK1 and STX17 using overexpression proteins to perform IP assay followed by WB with indicated antibodies.

Figure 6

Figure 6. Phosphorylation of ULK1 is degraded through chaperone-mediated autophagy. (d) The interaction of ULK1 to HSC70 was determined by pulldown assay with overexpression of GST-ULK1 and HA-HSC70.

-As mentioned, almost all work presented is based in protein-protein interaction, and always using an artificial overexpressed system. Endogenous conditions or alternative systems (PLA, colocalization) should be used to confirm that these interactions are physiologically relevant.

Response: Thanks for raising this issue. We agree with your point of view: Common techniques——IP/Co-IP/Pull-down assay can not discriminate between the direct and indirect protein-protein interaction. Consequently we used both endogenous conditions and PLA system to confirm the interaction between ULK1 and STX17. The endogenous binding assay is shown in **Fig. 4b** and **Fig.4c**. We also

use prokaryotic purification and pull down assay in vitro, we proved that the direct interaction between STX17 and ULK1 (**Fig.4f**).

We also performed PLA assay for different conditions such as serum starvation or ULK1 mimics in normal condition. All of these PLA related data are **Fig.4h, Fig.4i** and **Fig.4j**.

Figure 4

Figure 4

Figure 4. ULK1 physically interacts with STX17. (h) Myc-ULK1 or Myc vector were cotransfected with GFP-STX17. The PLA assay was performed and the signals (red) were detected under Confocal Microscopy. Scale Bar, 10 μ M.

Figure 4

Figure 4. ULK1 physically interacts with STX17. (i) The HeLa cells were transfected with GFP-STX17 and Myc-ULK1 with/out serum starvation (S.S). The PLA assay was performed with Myc and GFP antibodies. The fluorescence signal (red) was observed under Confocal Microscopy. Scale Bar, 10 μ M. Average of signal was quantified with a minimum of 20 cells. All values are means \pm SEM of at least three independent experiments. Student's t test (unpaired); *** $p < 0.001$.

Figure 4 j

Figure 4. ULK1 physically interacts with STX17. (j) The representative images of PLA assay of GFP-STX17 cotransfected with myc-ULK1 WT, S423A, S423D or myc-vector. Scale Bar, 10 μ M. Average of signal was quantified with a minimum of 20 cells. All values are means \pm SEM of at least three independent experiments. Student's t test (unpaired); ** $p < 0.01$, *** $p < 0.001$.

-The information given by authors regarding each experiment and including the main text, the figure legends, and material and methods is totally insufficient to understand each figure. There is a huge lack of details and information that make almost impossible to fully understand and identify all data presented. This is one of the main concerns of this manuscript.

Response: Thanks for this question. We lacked some experiments detail because the word numbers limitation of Nature Communication. We write the experimental details in this version to let audiences better understand our ideas.

-Almost all results are presented as a single image of a western blot. There is any quantification corresponding to different independent experiments to confirm that the changes observed are significant, especially when increase/decrease in the levels of proteins is the main result. Figure 3a is the only exception.

Response: Thanks for the question. We did the quantifications for our WB and Images. All the quantification data are inserted in the related figures.

SPECIFIC COMMENTS:

-Fig 1a: authors say that to determine the phosphorylation by PKC, different autophagy-related proteins were overexpressed, however

only results on ULK1 are presented. Did authors checked other proteins as they say or they directly checked ULK1? Why they say that “only the ULK1 showed a signal”?

Response: We do check several autophagy related proteins which are involved in key steps of autophagy. They are Vps34, BECN1, ULK1, ATG14 and ATG7. The figure 1a is replaced by this new figure (Fig 1) . The expressions of these proteins are presented in Fig. S1a.

Figure 1

Figure 1. S423 of ULK1 is the phosphorylation site by PKCα. (a) Phosphorylation of key autophagy related proteins were detected using p-PKC-Substrate antibody after immunoprecipitation. The expressions of these protein were shown in Fig. S1a.

Figure S1

Fig. S1 PKC phosphorylates ULK1 in vitro and in vivo. (a) The expressions of autophagy related key proteins and the specificities of IPs.

- Fig1a is impossible to understand what are the different lanes of the WB presented, what means "WLC"? What are the different bands detected in each lane and each blot? What type of sample is each lane? The information is missing in the main text and in the legend. This type of error is repeated all along the manuscript.

Response: It's typos of "WLC" in **Fig1a**. It should be "WCL" which means "whole cell lysate". We corrected it and other typos in the manuscript. Thank you for reminding us of this mistake.

As the suggestion, we also write the detail of each experiment through the manuscript.

- Fig1b: The p-PKC-substrate antibody is not specific to conclude that the kinase phosphorylating ULK1 is PKC-alpha, it can be any other PKC kinase.

Response: Thanks for the questions. It's true that we can't conclude that the phosphorylation kinase is PKC α only by using p-PKC-substrate antibody. We make this conclusion because we knockdown PKC α in U87 cells (**Fig. 1i**). In the U87 cells the PKC α is the dominant isoforms. This data is from Human Protein Atlas as **Fig S1e**. We also replaced the PKC α with PKC before **Fig.1b** in the manuscript.

Figure 1

Figure 1. S423 of ULK1 is the phosphorylation site by PKC α . (i) U87 cells were infected with virus of control or shPKC α s for 48 hrs. The lysates were analyzed with p-S423 antibody by WB. Quantification of p-S423. Bars are mean \pm SEM of triplicate samples, ** means $p < 0.01$, *** means $P < 0.001$. The efficiencies of shPKC α s were analyzed by PKC α antibody. Actin and PKC α were used as loading control.

Figure S1

Fig. S1 PKC phosphorylates ULK1 in vitro and in vivo. (e) The ratio of different PKC isoforms expression in U87 cells. The data is from Human Protein Atlas.

-Why in almost all experiments in this section, ULK1 proteins are overexpressed and immunoprecipitated against the same ULK1? Why changes in phosphorylation are not showed directly under endogenous levels without overexpression?

Response: Thanks for the concern. We tried to directly detect the endogenous ULK1 in many conditions. In most conditions, it is not easy detected. We thought the reason is that the p-S423 antibody is not so strong to detect less ULK1, this sort of situation for specific antibodies is very common¹. So we immunoprecipitated the total protein with endogenous ULK1 antibody (**Fig.1g,h,j**). Actually based on your concern, we optimized the usage of the p-423 antibodies and do determine the endogenous p-ULK1 S423 directly (**Fig.1i,S1.d,f**).

Figure 1

Figure 1. S423 of ULK1 is the phosphorylation site by PKCα. (g) The phosphorylation of endogenous ULK1 was detected by p-S423 antibody after HEK293T cells were transfected with PKCα WT or DN (dominant negative).

Figure 1

Figure 1. S423 of ULK1 is the phosphorylation site by PKC α . (h) The p-ULK1 S423 was analyzed by WB after HeLa cells were treated with PKC inhibitors GÖ6983 or Bis I for 0.5 h. p-PKC substrate antibody was used to determine the efficiency of drugs. Actin, ULK1 and PKC α were used as loading control.

Figure 1

Figure 1. S423 of ULK1 is the phosphorylation site by PKC α . (i) U87 cells were infected with virus of control or shPKC α s for 48 hrs. The lysates were analyzed with p-S423 antibody by WB. Quantification of p-S423. Bars are mean \pm SEM of triplicate samples, ** means $p < 0.01$, *** means $P < 0.001$. The efficiencies of shPKC α s were analyzed by PKC α antibody. Actin and PKC α were used as loading control.

-Fig S1c and others: control with empty Myc/Flag/HA-construct are always missing.

Response: Thanks for the suggestion. We redo the experiments with the empty vector control to make them better. Now it is **Fig. S2d**.

Figure S2

d

-Fig S1e and S1f are interchanged.

Response: Yes. We changed the **Fig S1e** and **S1f**. Now they are **Fig S2f** and **S2g**. Thank you for reminding us of this mistake.

-Fig S1f: why only fragment 2 pulldowns HA-ULK1? If the interaction region is between 167 and 338, fragment 4 or fragment 5 should also pulldown Flag-ULK1.

Response: Thank you for your question. The old **Fig S1f** is **Fig S2g** now. PKC α fragment 4 and fragment 5 both contain partial of fragment 2. We don't know the exact reason why only fragment 2 pulldown the ULK1. We think the most possibility is that this interaction needs the whole fragment 2. Both fragment 4 and 5 lacking partial of fragment 2 may change the conformation. We write them in manuscript from line 157 to 163 in page 6.

-FigS1i: in main text, the sentence is cut off?

Response: Thank you for your question. We rearranged our figures and article. Thanks again for your understanding.

-Fig2: All negative controls in all IPs are missing.

Response: We redo the experiments with all negative control in **Fig 2** and other Figs according to your suggestion. We changed with all new figures which contain negative control to make them better.

Figure 2

Figure 2. Phosphorylation of ULK1 doesn't affect ULLk1-ATG13 complex and its kinase activity. (a) Flag-ATG13 was transfected with Myc-ULK1 WT, S423A and S423D into HEK293T cells. The Ips were analyzed with Myc antibody. The densities of ATG13 were quantified. Bars mean \pm SEM of triplicate samples, NS: No Significance. **(b)** HA-FIP200 was transfected with Flag-ULK1 WT, S423A and S423D into HEK293T cells. IP with Myc antibody and analyzed with HA antibody. The densities of Fip200 were quantified. Bars mean \pm SEM of triplicate samples. NS: No Significance. **(c)** Flag-BECN1 was transfected with Myc-ULK1 WT, S423A and S423D or Myc-Control into HEK293T cells. IP with Myc antibody and analyzed with Flag antibody. The densities of BECN1 were quantified. Bars mean \pm SEM of triplicate samples. NS: No Significance. **(d)** Immunoprecipitation (IP) of ULK1 complex partners by Myc-ULK1 variants transfected into HEK293T cells and determined with ATG13, Fip200, BECN1 antibodies. The densities of ATG13, Fip200, BECN1 were quantified. Bars mean \pm SEM of triplicate samples. NS: No Significance.

-Fig2f: Staurosporine is a broad spectrum kinase inhibitor, is not specific for PKC-alpha, any other kinase (mTOR, AMPK...) might also be affected. Go6983 is a broad spectrum PKC inhibitor. Contrary to what authors say, P-Atg13 and P-Beclin1 are not decreased with the inhibitors, but P-mTOR does as well as p-PKC-sub. Authors don't explain the detail of what are the 5 types of different inhibitors used. The results in this experiment cannot support the conclusions.

Response: Thank you for your great suggestion. As suggested we don't use staurosporine as PKC inhibitor in the manuscript. Based on your and Reviewer 2's suggestions, we deleted all the panels which including staurosporine. Now we only use one inhibitor Go6983 in **fig.2f**. We modified this figure and the conclusion at line 203-209 in page 7.

Figure 2

Figure 2. Phosphorylation of ULK1 doesn't affect ULLk1-ATG13 complex and its kinase activity. (f) HeLa cells were treated with different concentrations of PKC inhibitor Gö6983 for 4 hrs. The lysates were analyzed with indicated antibodies. (g) The densities of p-ATG13, p-BECN1, p-ULK1 S757, p-mTORC1 were quantified. Bars mean \pm SEM of triplicate samples. ** $p < 0.01$; *** $p < 0.001$.

-Fig S2a/b: these results are inconclusive, cannot confirm that " PKC could regulate autophagy partially independent of mTOR" . The quality if fig S2a/ S2b is not good to say that the effect of both inhibitors is higher than with rapamycin alone.

Reponse: As mentioned above, we delete all the figures about staurosporine. Old **Fig S2a** was deleted. Old **Fig S2b** is **Fig S3a** now. We did more experiments to answer this question in **Fig S3b-S3e**. We also figure out the relation of PKC α and mTORC1 by knockdown PKC α or Raptor, mTor individually in **Fig S3c-S3e**. All of these data were presented from line 213 to 226 in page 8.

Figure S3

Supplemental Figure 3. PKC regulating autophagy is related to mTORC1. (a) HeLa cells were treated with PKC inhibitors combined with mTOR inhibitors for 0.5 h, autophagy was analyzed by WB with LC3 and p62 antibodies. The densities of LC-II and p62 were

quantified. All values are means \pm SEM of at least three independent experiments. Student's t test (unpaired); * $p < 0.05$, ** $p < 0.01$.

Figure S3

Supplemental Figure 3. PKC regulating autophagy is related to mTORC1. (b) The similar results were obtained in U87 cells as in S3a.

Figure S3

Supplemental Figure 3. PKC regulating autophagy is related to mTORC1. (e) The PKC α and mTORC1 activities were determined by p-PKC substrate and p-mTOR antibodies after PKC α was knocked down in HeLa. The mTOR activities were quantified. All values are means \pm SEM of at least three independent experiments. Student's t test (unpaired); * $p < 0.05$.

-Fig3: If ULK1- S423D (phospho mimic) is sent to degradation by CMA, why is it increasing autophagosome accumulation? Since p62 is not accumulated, we cannot attribute this effect to the inhibition of fusion. Authors do not give a clear hypothesis to explain why ULK1-S423D increase the number of autophagosome and this observation is quite spectacular.

Response: Thank you for your great question.

1) For your first concern: After the addition of CQ, S423A was more likely to accumulate P62 than WT and S423D, and we forgot to label the asterisk on the quantification of P62 in the old figure. As the figure orders have been relabeled (**Fig.3b, S4a**). Thanks for reminding us of this mistake.

2) For your second concern, it is a common mechanism of protein-Protein regulation that "A" protein binding to "B" Protein to promote or attenuate the interaction between "B" protein and "C" protein. For examples, Smad7 enhances the interaction of YY1 with the histone deacetylase HDAC1²; HSC90 α ¹ promotes the binding of PKM2 to Bcl2³; 14-3-3 binding to p-BMAL to inhibit its interaction to CLOCK⁴. Besides, a protein or a protein complex regulates both initiation and late stages of autophagy have been reported, such as Atg14L⁵ and SMCR8/C9ORF72⁶.

Our model is: STX17 and Snap29 participate in the fusion of autophagosome and lysosome^{7, 8}. We found that the interaction between STX17 and Snap29 is mediated by ULK1. S423A or non-P-S423-ULK1 enhanced the STX17 and Snap29's interaction (**Fig.4g, Fig.5a,b,g,h**). STX17 and Snap29's interaction requires the binding between ULK1 and STX17. Overexpression of Snap29 decrease the binding between ULK1 and STX17 (**Fig.S7b**). Knockdown of Snap29 increase the binding between ULK1 and STX17 (**Fig.S7c**). In conclusion, after enhancement of STX17 and Snap29' s interaction, non-p-ULK1 or S423A is dissociated from STX17 and degraded through preteasome pathway (**Fig.6a,b**).

If ULK1 does not mediate the enhancement process, STX17 lost the ability to interact with Snap29 better, and the fusion process can not be carried out smoothly. P-ULK1 has a low affinity to STX17, and

is degraded through CMA. Consequently P-ULK1 does not promote the interaction between STX17 and Snap29, then the fusion process is indolent, which causes the accumulation of autophagosomes.

Figure 3

b

Figure 3. Phosphorylation of ULK1 play a key role in fusion of autophagosome to lysosome. (b) The quantified ratio (CQ-ULK1s/ULK1s) of LC3-Π and p62 were shown. All values are means ± SEM of at least three independent experiments. Student's t test (unpaired); **p< 0.01, ***p< 0.001.

Figure S4

a

Supplemental Figure 4. Phosphorylation of ULK1 regulates autophagosome fusion. (a) WB analysis of LC3 and p62 in ULK1 WT and mutants overexpressing cells with/out CQ treatment are represented as

quantification of LC3-Π and p62 intensity (relative to actin).

Figure 4

Figure 4. ULK1 physically interacts with STX17. (g) The mice were injected with GÖ6983 or PBS for 7 days via tail vein. The livers lysates were performed IP with STX17 antibody. The IPs were detected by ULK1 antibody. The indicated antibodies were used to confirm the drug efficiency.

Figure 5

Figure 5. Phosphorylation of ULK1 influences STX17-SNAP29-VAMP8 complex. (a) The HeLa cells were starved with time course and the lysates were performed with Ip assay via STX17 antibody. The IPs were analyzed with ULK1 antibody. The IPed ULK1 was quantified. All values are means \pm SEM of at least three independent experiments. Student's t test (unpaired); ** $p < 0.01$.

Figure 5

Figure 5. Phosphorylation of ULK1 influences STX17-SNAP29-VAMP8 complex. (b) ULK1 S423D decreased binding to STX17 by IP assay. And the IPed STX17 was quantified. All values are means \pm SEM of at least three independent experiments. Student's t test (unpaired); * $p < 0.05$, ** $p < 0.01$.

Figure 5

Figure 5. Phosphorylation of ULK1 influences STX17-SNAP29-VAMP8 complex. (g) The myc-ULK1 variants transfected HeLa cells were treated with CQ or PBS. The IP assay was performed with STX17 antibody or IgG. and detected by indicated antibodies. The IP bands were quantified. The IP bands were quantified. All values are means \pm SEM of at least three independent experiments. Student's t test (unpaired); * $p < 0.05$, ** $p < 0.01$, *** $p < 0.001$.

Figure 5

Figure 5. Phosphorylation of ULK1 influences STX17-SNAP29-VAMP8 complex. (h) The lysates from the livers of the fasting or normal chew mice were performed IP assay with STX17 antibody and analyzed with ULK1 antibody.

Figure S7

Supplemental Figure 7. Phosphorylation of ULK1 regulates STX17/SNAP29 complex. (b) Overexpression of SNAP29 decreases STX17 binding ULK1. GST-ULK1 WT, S423A were cotransfected with GFP-SNAP29 or GFP into HEK293T cells. The pull-down was detected by STX17 antibody. The IPed STX17 was quantified. All values are means \pm SEM of at least three independent experiments. Student's t test (unpaired); * $p < 0.05$, *** $p < 0.001$.

Figure S7

c

values are means \pm SEM of at least three independent experiments. Student's t test (unpaired); * $p < 0.05$, ** $p < 0.01$.

Supplemental Figure 7.

Phosphorylation of ULK1 regulates STX17/SNAP29 complex.

(c) Knockdown of SNAP29 increases STX17 and ULK1 affinity. GST-ULK1 WT or S423A was cotransfected with FB-STX17 into HEK293 cells, then infected with ShRNAs-SNAP29 lentivirus or control lentivirus. The pulldown was detected by Flag antibody. The IPed STX17 was quantified. All

Figure 6

Figure 6. Phosphorylation of ULK1 is degraded through chaperone-mediated autophagy. (a)

MG132 (proteasome inhibitor) prevents ULK1 S423A degradation rather than ULK1 WT and S423D. Stable ULK1 WT and

mutant HeLa cells were pretreated Cycloheximide (CHX), an inhibitor of protein synthesis, for 3 hours. Then the cells were treated with MG132 for time course with the presence of CHX. The remained ULK1 were detected by WB. (b) Chloroquine (CQ, an inhibitor of lysosome) inhibits ULK1 S423D degradation. The procedure is similar to Fig. 6a but MG132 is replaced by CQ.

-Fig 4 and 5: same deficiencies as in previous figures, controls are missing and information to understand the details of the experiments are lacking.

Response: Thank you for raising this issue. As suggested, we redo or relabel the experiments with negative controls and annotations such as in **Fig 4a-4c 4f** and **Fig 5b,5g**.

Figure 4

Figure 4. ULK1 physically interacts with STX17. (a,b,c) Interaction between ULK1 and STX17 using overexpression (a) and endogenous (b,c) proteins to perform IP assay followed by WB

with indicated antibodies. (f) The GST-ULK1 279-525 and His-STX17 157-275 were purified from bacteria and incubated with Glutathione beads for 4 hrs. After washing, the elution was analyzed with His antibody.

Figure 4

Figure 4. ULK1 physically interacts with STX17. (f) The GST-ULK1 279-525 and His-STX17 157-275 were purified from bacteria and incubated with Glutathione beads for 4 hrs. After washing, the elution was analyzed with His antibody.

Figure 5

Figure 5. Phosphorylation of ULK1 influences STX17-SNAP29-VAMP8 complex. (b) ULK1 S423D decreased binding to STX17 by IP assay. And the IPed STX17 was quantified. All values are means \pm SEM of at least three independent experiments. Student's t test (unpaired); * $p < 0.05$, ** $p < 0.01$.

Figure 5

-Fig 6a: authors say that MG132 can inhibit degradation of ULK1 WT and S423A by UPS but not S423D. However Fig 6a shows that UPS-dependent degradation prevented by MG132 treatment occurs only with S423A, but not WT or S423D.

Response: This is a great question, thanks. We say that unphosphorylated ULK1 S423 is degraded by UPS but phosphorylated ULK1 is degraded by CMA. In **Fig 6a**, MG132 treatment does not prevent ULK1 WT degradation. We believe that in the cell phosphorylation and unphosphorylation of a protein are balanced. WT could be switched between phosphorylation and unphosphorylation status. As in **Fig 6a** ULK1 WT is also in the balance. The p-ULK1 is degraded and the unphosphorylated form will be phosphorylated to make the balance so the total ULK1 WT is

reduced. Here our conclusion is not accurate. We change this in the manuscript and draw the experimental schematic for **Fig.6a, b** (**Fig.S9a,b**). Thank you again for raising the great question.

Figure S9

Supplemental Figure 9. ULK1 is linked CMA to macroautophagy. (a) CHX and MG132 were used to inhibit the translation process and proteasome, ULK1s' degradations through lysosome were detected by WB. (b) CHX and CQ were used to inhibit the translation process and lysosome, ULK1s' degradations through proteasome were detected by WB.

Figure 6

Figure 6. Phosphorylation of ULK1 is degraded through chaperone-mediated autophagy. (a) MG132 (proteasome inhibitor) prevents ULK1 S423A degradation rather than ULK1 WT and S423D. Stable ULK1 WT and mutant HeLa cells were pretreated with Cycloheximide (CHX), an inhibitor of protein synthesis, for 3 hours. Then the cells were treated with MG132 for a time course with the presence of CHX. The remaining ULK1 was detected by WB. (b) Chloroquine (CQ, a lysosome inhibitor) inhibits ULK1 S423D degradation. The procedure is similar to Fig. 6a but MG132 is replaced by CQ.

-CMA degradation is mediated by isoform LAMP-2A. Knock-out cells used to study CMA process should only delete this isoform but not LAMP-2B and LAMP2C that might have other important roles in lysosomal biology.

Response: We thanks for the concern. It's true that CMA degradation is mediated by isoform LAMP2a. As suggested we knockdown LAMP2a isoform by pGPIZ lentivirus system in HeLa cells. We detected the ULK1 degradation in this cell line as in **Fig S8a** which is in the manuscript of line 415-417 in page 14.

Figure S8

Supplemental Figure 8. ULK1 is degraded through CMA. (a) The LAMP2 was knockout through CRISPR/Cas9 system and LAMP2a was knockdown by pGPIZ lentivirus system in HeLa cells. The ULK1 and PKM2 were determined by WB. The quantification was calculated and all values are means \pm SEM of at least three independent experiments. Student's t test (unpaired); **p<0.01, ***p<0.001.

-Fig 6c: other typical CMA substrates should be showed here as a positive control to confirm that this slight increase in ULK1 levels when total LAMP-2 is missing, is due to a lack of ULK1 degradation by CMA.

Response: As suggested we detected another CMA substrate PKM2 as positive control with ULK1 in LAMP2a knockdown cells as in **Fig S8a**⁹.

Figure S8

Reviewer #2 (Remarks to the Author):

In the manuscript by Wang et al., the authors investigated the role of ULK1 for a later step of the autophagy signaling process, i.e. the fusion of autophagosomes with lysosomes. The authors describe that ULK1 can be phosphorylated by PKCa at Ser423. This phosphorylation reduces the affinity of ULK1 to STX17. In contrast, the authors suggest that unphosphorylated ULK1 recruits STX17 to the autophagosome membrane, where it has a higher affinity for SNAP29. Phosphorylated ULK1 binds to HSC70 and becomes degraded through chaperone-mediated autophagy. Collectively, this study identifies ULK1 as an important regulator of late steps of the autophagy process. Although this is a very interesting study, I think major revisions are necessary in order to make this work acceptable for publication in NATURE COMMUNICATIONS. I think the authors need to address the following aspects.

Major points:

1) I think the authors need to better explain how PKC-dependent phosphorylation of ULK1 at Ser423 is regulated during autophagy. It appears that under nutrient-rich conditions, this site is phosphorylated (Figure 1I).

a. Does this in turn mean that ULK1 becomes constitutively degraded by CMA? According to figures 6A and 6B, ULK1 degradation is blocked neither by MG132 nor by chloroquine.

Response: Thanks for the question. We observed that phosphorylated ULK1 is degraded by CMA but unphosphorylated form is degraded through proteasome. So ULK1 WT may be degraded through both CMA and proteasome because ULK1 WT could switch from unphosphorylated form to phosphorylated form. That is why ULK1 WT degradation is blocked neither by MG132 nor by chloroquine.

b. Does autophagy induction lead to PKC inactivation? Or does autophagy activate a specific phosphatase dephosphorylating this site?

Response: We are not sure whether autophagy induction leads to PKC inactivation. But we have data to show that reduction of mTORC1 activity which stimulates autophagy also decrease PKC activity as shown in **Fig S3c-S3d**. Actually knockdown of PKC α also decreases mTORC1 activity (**Fig. S3e**). We showed the data in the manuscript from line 220 to 225 in page 8.

Figure S3

Supplemental Figure 3. PKC regulating autophagy is related to mTORC1. (c,d) The PKC α and mTORC1 activities were determined by p-PKC substrate and p-4EBP antibodies after Raptor or mTOR was knocked down in both HeLa and HEK293 cells. The PKC activities were quantified. All values are means \pm SEM of at least three independent experiments. Student's t test (unpaired); ** $p < 0.01$, *** $p < 0.001$. (e) The PKC α and mTORC1 activities were determined by p-PKC substrate and p-mTOR antibodies after PKC α was knocked down in HeLa. The mTOR activities were quantified. All values are means \pm SEM of at least three independent experiments. Student's t test (unpaired); * $p < 0.05$.

c. I think the usage of staurosporine as PKC inhibitor should be avoided. This compound inhibits several members of the human kinome. (staurosporine deleted)

Response: Thank you for your advice. As suggested, we deleted all figures and panels including staurosporine.

d. Apparently, ULK1 Ser423 phosphorylation is oscillating (Figure 1I). Does this mean that PKC becomes reactivated after 1.5 h of serum starvation?

Response: Thanks for the question.

We knew that mTORC1 is being reactivated during serum starvation¹⁰. From our data in **Fig.S3c-e**, the mTORC1 and PKC α could affect each other. So PKC reactivation is possible.

As we know, autophagy also displays oscillation during the time course of serum starvation¹¹. When autophagy level rise to the first peak after 1 hour of serum starvation¹¹, PKC activity is the lowest (**Fig.1i, S7a**). Autophagy level gradually decreased between 1-2 hours¹¹. Consequently PKC reactivation after 1.5 h of serum starvation consistent with the autophagy is possible. We have shown that ULK1 is the substrate of PKC α (**Fig.1, S1, S2**), and PKC α is dominant in HeLa or U87 cells (**Fig. 1i, S1d, e**). So the level of P-423 represents the activity of PKC largely (**Fig.1j**). It means that PKC becomes reactivated after 1.5 h of serum starvation.

Figure S3

Supplemental Figure 3. PKC regulating autophagy is related to mTORC1. (c,d) The PKC α and mTORC1 activities were determined by p-PKC substrate and p-4EBP antibodies after Raptor or mTOR was knocked down in both HeLa and HEK293 cells. The PKC activities were quantified. All values are means \pm SEM of at least three independent experiments. Student's t test (unpaired); ** $p < 0.01$, *** $p < 0.001$. (e) The PKC α and mTORC1 activities were determined by p-PKC substrate and p-mTOR antibodies after PKC α was knocked down in HeLa. The mTOR activities were quantified. All values are means \pm SEM of at least three independent experiments. Student's t test (unpaired); * $p < 0.05$.

Figure 1

Figure 1. S423 of ULK1 is the phosphorylation site by PKC α . (i) U87 cells were infected with virus of control or shPKC α s for 48 hrs. The lysates were analyzed with p-S423 antibody by WB. Quantification of p-S423. Bars are mean \pm SEM of triplicate samples, ** means $p < 0.01$, *** means $P < 0.001$. The efficiencies of shPKC α s were analyzed by PKC α antibody. Actin and PKC α were used as loading control.

Figure S7

Supplemental Figure 7. Phosphorylation of ULK1 regulates STX17/SNAP29 complex. (a) The GST-ULK1 was cotransfected with GFP-STX17 and followed with serum starvation for different times. The Glutathione pulldown was analyzed with GFP antibody. The IPed GFP-STX17 was quantified. All values are means \pm SEM of at least three independent experiments. Student's t test (unpaired); *** $p < 0.001$.

Figure S1

Fig. S1 PKC phosphorylates ULK1 in vitro and in vivo. (d) HeLa cells were infected with viruses of control or shPKC α s for 48 hrs. The lysates were analyzed with p-S423 antibody by WB. The efficiencies of shPKC α s were analyzed by PKC α antibody. Actin and PKC α were used as loading control.

Figure 1

Figure 1. S423 of ULK1 is the phosphorylation site by PKC α . (j) p-S423 was determined during serum starvation by WB. p-S423 was consistent with PKC α activity. The p-S423 was quantified. Bars are mean \pm SEM of triplicate samples, ** means $p < 0.01$

2) To this reviewer, it is not clear how non-phosphorylated ULK1 regulates the interaction between STX17 and SNAP29. In figure 5G, it appears that SNAP29 does not affect the interaction between ULK1 S423A and STX17. However, the interaction between wt ULK1 and STX17 can be reduced by SNAP29 overexpression. How can the authors explain this?

Response: Thank you for your question. The formation of autolysosome is mediated by the interaction of STX17 and Snap29^{7, 8}. We found that the interaction is enhanced by S423A or non-P-S423-ULK1 (**Fig.4g, Fig.5a,b,g,h**). We did the quantifications for our result of IP in **Fig 5G** (now it is **Fig.S7b**), we found both WT and S423A had a significant difference. S423A binds to STX17 stronger than WT. Overexpression of Snap29 weakens the interaction between STX17 and S423A, but the decline (IPed STX17) is not so obvious as WT after GFP-Snap29 overexpression. The most possible reason is that S423A is a permanent and rigid mutation or mimic.

Overexpression of Snap29 decreases the binding between ULK1 and STX17 (**Fig.S7b**). Knockdown of Snap29 increases the binding between ULK1 and STX17 (**Fig.S7c**). In conclusion, after

enhancement of STX17 and Snap29' s interaction transiently ULK1 is dissociated from STX17 and degraded through preteasome mediated pathway(Fig.6a,b,S9a,b).

Figure 4

Figure 4. ULK1 physically interacts with STX17. (g) The mice were injected with GÖ6983 or PBS for 7 days via tail vein. The livers lysates were performed IP with STX17 antibody. The IPs were detected by ULK1 antibody. The indicated antibodies were used to confirm the drug efficiency.

Figure 5

Figure 5. Phosphorylation of ULK1 influences STX17-SNAP29-VAMP8 complex. (a) The HeLa cells were starved with time course and the lysates were performed with Ip assay

via STX17 antibody. The IPs were analyzed with ULK1 antibody. The IPed ULK1 was quantified. All values are means \pm SEM of at least three independent experiments. Student's t test (unpaired); ** $p < 0.01$. (b) ULK1 S423D decreased binding to STX17 by IP assay. And the IPed STX17 was quantified. All values are means \pm SEM of at least three independent experiments. Student's t test (unpaired); * $p < 0.05$, ** $p < 0.01$. (g) The myc-ULK1 variants transfected HeLa cells were treated with CQ or PBS. The IP assay was performed with STX17 antibody or IgG. and detected by indicated antibodies. The IP bands were quantified. The IP bands were quantified. All values are means \pm SEM of at least three independent experiments. Student's t test (unpaired); * $p < 0.05$, ** $p < 0.01$, *** $p < 0.001$. (h) The lysates from the livers of the fasting or normal chew mice were performed IP assay with STX17 antibody and analyzed with ULK1 antibody.

Figure S7

Supplemental Figure 7. Phosphorylation of ULK1 regulates STX17/SNAP29 complex.
(b) Overexpression of SNAP29 decreases STX17 binding ULK1. GST-ULK1 WT, S423A were cotransfected with GFP-SNAP29 or GFP into HEK293T cells. The pull-down was detected by STX17 antibody. The IPed STX17 was quantified. All values are means \pm SEM of at least three independent experiments. Student's t test (unpaired); * $p < 0.05$, *** $p < 0.001$.
(c) Knockdown of SNAP29 increases STX17 and ULK1 affinity. GST-ULK1 WT or S423A was cotransfected with FB-STX17 into HEK293 cells, then infected with ShRNAs-SNAP29 lentivirus or control lentivirus. The pull-down was detected by Flag antibody. The IPed

Figure 6. Phosphorylation of ULK1 is degraded through chaperone-mediated autophagy. (a) MG132 (proteasome inhibitor) prevents ULK1 S423A degradation rather than ULK1 WT and S423D. Stable ULK1 WT and mutant HeLa cells were pretreated with Cycloheximide (CHX), an inhibitor of protein synthesis, for 3 hours. Then the cells were treated with MG132 for a time course with the presence of CHX. The remaining ULK1 was detected by WB. (b) Chloroquine (CQ, an inhibitor of lysosome) inhibits ULK1 S423D degradation. The procedure is similar to Fig. 6a but MG132 is replaced by CQ.

3) The authors only use serum starvation in figures 1I and 5A, and in figure 5I livers of fasting mice were used. With regard to the analysis of colocalization between ULK1 and STX17 on the one hand and between ULK1 and LAMP2 on the other hand by immunofluorescence, it would be interesting to perform these analyses under serum starvation.

Response: Thank you for your great suggestions. As suggested we performed the immunofluorescence of ULK1 and Lamp2 or LAMP2a under serum starvation condition and quantified the colocalization. The colocalization is lower in serum starvation (**Fig S8b, S8c**). This is in the manuscript from line 417 to 419, page 14. We also performed PLA assay to analyze the colocalization of ULK1 and STX17 under serum starvation. It clearly showed that the colocalization is higher in serum starvation (**Fig 4i**). This data was presented in manuscript from line 306 to 310, page 11.

Figure S8

Supplemental Figure 8. ULK1 is degraded through CMA. (b) Representative images of HeLa cells immunostaining with GFP-ULK1 and LAMP2. The cells were performed immunostaining for LAMP2 (Red) and GFP (Green) and taken the images under confocal

microscope. Scale Bar, 5 μ M. The represented figure as quantification from a minimum of 20 cells. All values are means \pm SEM of at least three independent experiments. Student's t test (unpaired); ** $p < 0.001$. (c) Representative images of HeLa cells immunostaining with Myc-ULK1 and LAMP2a. The cells were performed immunostaining for LAMP2a (Red) and Myc (Green) and taken the images under confocal microscope. Scale Bar, 5 μ M. The represented figure as quantification from a minimum of 20 cells. All values are means \pm SEM of at least three independent experiments. Student's t test (unpaired); ** $p < 0.001$.

Figure 4

Figure 4. ULK1 physically interacts with STX17. (i) The HeLa cells were transfected with GFP-STX17 and Myc-ULK1 with/out serum starvation (S.S). The PLA assay was performed with Myc and GFP antibodies. The fluorescence signal (red) was observed under Confocal Microscopy. Scale Bar, 10 μ M. Average of signal was quantified with a minimum of 20 cells. All values are means \pm SEM of at least three independent experiments. Student's t test (unpaired); *** $p < 0.001$.

Minor points:

1) The manuscript needs considerable proofreading. At present, the usage of the English language is not appropriate. For example line 242: "Partial of STX17 colocalized with ATG16 except STX17." (but there are several more passages)

Response: As suggested we proofread carefully and modified them including "Partial of STX17 colocalized with ATG16 except STX17."

2) Sometimes, the statements in the manuscript text do not reflect what can be seen in the figures:

a. Lines 160-162: The authors describe that phosphorylation of ATG13 and Beclin 1 is decreased after treatment with PKC inhibitors. In figure 2F, the signals appear to be increased. Furthermore, this indicates that ULK1 phosphorylation at Ser423 does affect ULK1 kinase activity, although the authors state that this is not the case.

Response: Thanks for the question. We proofread again and modified these descriptions of **Fig.2f,g**. We deleted the lanes of staurosporine treatment. mTORC1 activity is decreased by the PKC inhibitor too. So p-ATG13 deduction may be due to p-ULK1 S757 by mTORC1.

Figure 2

Figure 2. Phosphorylation of ULK1 doesn't affect ULLK1-ATG13 complex and its kinase activity. (f) HeLa cells were treated with different concentrations of PKC inhibitor GÖ6983 for 4 hrs. The lysates were analyzed with indicated antibodies.

b. Lines 188-190: the calculations of the ratios are not reflected by the diagrams shown in figure 3A

Response: Thanks for your question. We have considered your opinion and added the figures (**Fig.3b,d**). For the obtention of the ratios of the results, we described them in the manuscript line 237-241, line 246-249, page 8.

Figure 3

Figure 3. Phosphorylation of ULK1 play a key role in fusion of autophagosome to lysosome. (b) The quantified ratio (CQ-ULK1s/ULK1s) of LC3-II and p62 were shown. All values are means \pm SEM of at least three independent experiments. Student's t test (unpaired); ** $p < 0.01$, *** $p < 0.001$. **(d)** The quantificated ratio(CQ-ULK1s/ULK1s) of puncta per cell were shown. All values are means \pm SEM of at least three independent experiments. Student's t test (unpaired); *** $p < 0.001$.

c. Line 282: "MG132 could prevent WT and S423A degradation". In figure 6A, WT degradation is clearly not prevented.

Response: Thanks for the concern. We modified them in the manuscript line 371-373, page 13.

3) In some experiments, the authors make use of ULK1/2 KO HeLa cells. However, the authors do mention the origin of this cell line. I assume they were generated by CRISPR/Cas9, but in the Materials & Methods section only sgULK1 is mentioned. The authors should clearly indicate how this cell line was generated. Furthermore, the usage of this cell line should be indicated in the individual figures.

Response: Thank you for your advice. The details of ULK1/2 DKO Hela cells have been put in the experimental materials and methods.

4) I do not request the quantification of all western blots. However, I think in some cases quantification is mandatory (e.g. figures 1I, 2D-F, supplemental figures S1C, S2, etc.)

Response: As suggested, we quantified most western blots including **figures 1I, 2D-F, supplemental figures S1C, S2.**

5) Figure 1A: The different lanes need to be labeled.

Response: We replaced the **Fig 1a** as requested by another reviewer. Thanks for your understanding.

6) Figure 1E: "1st" and "2nd" need to be explained in the legend.

Response: As suggested we rearranged the panels and explained the specific procedure of the experiment carefully of new **Fig 1f** (referred as old **Fig 1e**). Thanks.

Figure 1

Figure 1. S423 of ULK1 is the phosphorylation site by PKC α . (f) Phosphorylation site was confirmed by in vitro kinase assay with full length ULK1. The indicated GST tagged ULK1s were transfected into HEK293T cells, Followed by GST pull down(Step1), CIP the indicated Beads(Step2), PKC α Kinase assay(Step3), and CIP the whole PVDF membrane of the westernblot to confirm the phosphorylation(Step4).

7) Figure 3C: The red stars are almost not visible.

Response: As requested we move this figure to **Fig S4c**. We use the large images to show the detail.

Figure S4

Supplemental Figure 4. Phosphorylation of ULK1 regulates autophagosome fusion. (c) Accumulation of complete double membrane autophagosome (red star) by TEM in ULK1 mutants stable cells in ULK1-KO cells with/out CQ treatment (n=15). N: Nuclei.

8) Figure 4: The authors identified aa 157-275 of STX17 and aa 279-525 of ULK1 responsible for the interaction. Can the authors comment why aa 1-216 of STX17 and aa 17-465 of ULK1 were used for the in vitro pull down experiment shown in figure 4F?

Response: Thanks for the question. We agree with your opinion: aa 1-216 of STX17 and aa 17-465 of ULK1 were not accurate for the in vitro pull down experiment. We purified the fragment 6* His STX17 157-275 and fragment GST-ULK1 279-525 from bacteria and performed in vitro binding assay. The result is showed as **Fig. 4f** instead of previous figure.

Figure 4

Figure 4. ULK1 physically interacts with STX17. (f) The GST-ULK1 279-525 and His-STX17 157-275 were purified from bacteria and incubated with Glutathione beads for 4 hrs. After washing, the elution was analyzed with His antibody.

9) Supplemental Figure S1: From panel S1C, different binding affinities between WT/S423A/S423D and PKC are not evident (see minor point 4).

Response: We redo this experiment and the data is **Fig. S2d** now. We quantified the IPed PKC α and the S423D significantly bond more PKC α .

Figure S2

Supplemental Figure 2. PKC α physically interacts with ULK1. (d) FB-ULK1 variants or FB vector transfected into HEK293T cells, and the lysates were IP with SBP beads and then detected with PKC α antibody. The IPed PKC α were quantified. All values are means \pm SEM of at least three independent experiments. Student's t test (unpaired); * **p < 0.01.

10) Supplemental Figure S1: I think labeling of "e" and "f" has to be switched.

Response: As suggested we switched "e" and "f" in the manuscript. Thank you for reminding us of this mistake.

11) Supplemental Figure S2: Here, the authors did not analyze autophagic flux (+/- BafA1 or CQ). Furthermore, quantification of this figure is essential (see minor point 4), since the difference between PKC/mTOR inhibition vs. mTOR inhibition alone is not obvious.

Response: As reviewers suggested we performed the experiment with CQ to analyze autophagic flux (**Fig. S3b**). We quantified the **Fig.S3a** (previous **Fig. S2b**) and the difference between PKC/mTOR inhibition vs mTOR inhibition alone is obvious. Thank you for your suggestion.

Figure S3

Supplemental Figure 3. PKC regulating autophagy is related to mTORC1. (a) HeLa cells were treated with PKC inhibitors combined with mTOR inhibitors for 0.5 h, autophagy was analyzed by WB with LC3 and p62 antibodies. The densities of LC-II and p62 were quantified. All values are means \pm SEM of at least three independent experiments. Student's t test (unpaired); * $p < 0.05$, ** $p < 0.01$. (b) The similar results were obtained in U87 cells as in S3a.

12) Supplemental Figure S4: The authors just present a workflow and a silver staining. There are no indications how mass spectrometry was performed.

Response: Thank you for raising this issue. As suggested we added the procedure of LC-MS/MS technology to the Methods and Materials.

13) Supplemental Figure S6: The authors do not comment on the reduced co-localization between ULK1 S423A and LAMP2, although this does support their conclusion.

Response: Thank you for raising this issue. We added the related comment in the manuscript line 329-332, page11-12.

Reviewer #3 (Remarks to the Author):

There is considerable interest for the mechanisms controlling the biogenesis of autophagosomes. In this manuscript, authors report an interesting role of ULK1 protein in regulating SNARE-mediated membrane fusion for autophagosome maturation. This new role of ULK1 protein is modulated by the phosphorylation at S423 via PKC α . However, I would not recommend publication in Nature Communications.

First of all, ULK1 is a well-known marker for omegasome/phagophore before autophagosome formation. For example, a previous study suggested that ULK1 dissociates from the phagophore membrane together with other early ATG proteins [Autophagy 9, 1491, 2013]. How can ULK1 regulate SNARE-mediated autophagosome/lysosome fusion even though ULK1 is not on autophagosomes?

Response: Thanks for your great concern. We would like to address these issues by following discussions.

1) It is a common mechanism of protein-Protein regulation that "A" protein binding to "B" Protein to promote the interaction between "B" protein and "C"protein. For examples, Smad7 enhances the interaction of YY1 with the histone deacetylase HDAC1²; HSC90 α ¹ promotes the binding of PKM2 to Bcl2³. Besides, a protein or a protein complex regulate both initiation and late stages of autophagy has been reported, such as Atg14L⁵ and SMCR8/C9ORF72⁶.

Based on the experiments, our model is: STX17 and the Snap29, promote the formation of autolysosome^{7, 8}, we found that the interaction between STX17 and Snap29 is mediated by ULK1. S423A or non-P-S423 ULK1 enhanced the STX17 and Snap29's interaction (**Fig.4g, Fig.5a,b,g,h**). In this process, S423A-ULK1 and LC3 have better colocalization which strongly indicate that S423A-ULK1 involved in the late stage of autophagy (**Fig.5c**). Overexpression of Snap29 decreases the binding between ULK1 and STX17 ((**Fig. S7b**)). Knockdown of Snap29 increases the binding between ULK1 and STX17 (**Fig. S7c**). In conclusion, after enhancement of STX17 and Snap29's interaction transiently, ULK1 is dissociate from STX17 and degraded through preteasome mediated pathway (**Fig.6a,b**).

2) If the S423A or non-P-ULK1 remains on the membrane of the autophagosome all the time, after the enhancement of STX17/Snap29/Vamp8 formation, S423A or non-P-ULK1 will go with STX17/Snap29/Vamp8 to the lysosome pathway for degradation, rather than to the proteasome.

We overexpressed ULK1 S423D in ULK1/2 DKO cells and saw the autophagosome is accumulated in these cells and there are no changes in LC3-II and puncta with/out CQ treatment (**Fig 3a, 3c, S4c**). We concluded that ULK1 plays a role in autophagosome and lysosome fusion. Furthermore we observed that ULK1 S423A is colocalized with LC3 on the puncta largely (**Fig.5c**). It means that ULK1 does exit on autophagosome. It's possible that most ULK1 dissociates from the phagophore membrane together with other early ATG proteins, however, partial ULK1 still remains on phagophore membrane during autophagosome mature. The ULK1 dissociates from autophagosome after SNAP29 interacts with STX17.

Thanks again for your great question.

Figure 4

Figure 4. ULK1 physically interacts with STX17. (g) The mice were injected with GÖ6983 or PBS for 7 days via tail vein. The livers lysates were performed IP with STX17 antibody. The IPs were detected by ULK1 antibody. The indicated antibodies were used to confirm the drug efficiency.

Figure 5

Figure 5. Phosphorylation of ULK1 influences STX17-SNAP29-VAMP8 complex. (a) The HeLa cells were starved with time course and the lysates were performed with Ip assay via STX17 antibody. The IPs were analyzed with ULK1 antibody. The IPed ULK1 was quantified. All values are means \pm SEM of at least three independent experiments. Student's t test (unpaired); ** $p < 0.01$. **(b)** ULK1 S423D decreased binding to STX17 by IP assay. And the IPed STX17 was quantified. All values are means \pm SEM of at least three independent experiments. Student's t test (unpaired); * $p < 0.05$, ** $p < 0.01$. **(g)** The myc-ULK1 variants transfected HeLa cells were treated with CQ or PBS. The IP assay was performed with STX17 antibody or IgG. and detected by indicated antibodies. The IP bands were quantified. The IP bands were quantified. All values are means \pm SEM of at least three independent experiments. Student's t test (unpaired); *, $p < 0.05$,** $p < 0.01$,*** $p < 0.001$. **(h)** The lysates from the livers of the fasting or normal chew mice were performed IP assay with STX17 antibody and analyzed with ULK1 antibody.

Figure 5

Figure 5. Phosphorylation of ULK1 influences STX17-SNAP29-VAMP8 complex. (c,d) GFP-STX17, Flag-LC3 and Myc-ULK1 variants were cotransfected and detected by immunostaining in HeLa cells (n=20) and the quantification of images (d). Scale bars, 10 μm. All values are means ± SEM of at least three independent experiments. Student's t test (unpaired); ***p< 0.001.

Figure S7

c

Supplemental Figure 7. Phosphorylation of ULK1 regulates STX17/SNAP29 complex. (b) Overexpression of SNAP29 decreases STX17 binding ULK1. GST-ULK1 WT, S423A were cotransfected with GFP-SNAP29 or GFP into HEK293T cells. The pull-down was detected by STX17 antibody. The IPed STX17 was quantified. All values are means \pm SEM of at least three independent experiments. Student's t test (unpaired); * $p < 0.05$, *** $p < 0.001$. (c) Knockdown of SNAP29 increases STX17 and ULK1 affinity. GST-ULK1 WT or S423A was cotransfected with FB-STX17 into HEK293 cells, then infected with ShRNAs-SNAP29 lentivirus or control lentivirus. The pull-down was detected by Flag antibody. The IPed STX17 was quantified. All values are means \pm SEM of at least three independent experiments. Student's t test (unpaired); * $p < 0.05$, ** $p < 0.01$.

Figure 6

Figure 6. Phosphorylation of ULK1 is degraded through chaperone-mediated autophagy. (a) MG132(proteasome inhibitor) prevents ULK1 S423A degradation rather than ULK1 WT and S423D. Stable ULK1 WT and mutant HeLa cells were pretreated Cycloheximide(CHX), a inhibitor of protein synthesis, for 3 hours. Then the cells were treated with MG132 for time course with the presence of CHX. The remained ULK1 were detected by WB. (b) Chloroquine(CQ, a inhibitor of lysosome) inhibits ULK1 S423D degradation. The procedure is similar to Fig. 6a but MG132 is replaced by CQ.

Secondly, this paper is not convincing. The major claims of this manuscript are not fully supported by direct evidences.

1. In Figure 5, since authors only tested the interaction between STX17 and SNAP29 or STX17 and VAMP8, it is hard for them to claim the influence on STX17-SNAP29-VAMP8 complex formation requiring three proteins simultaneously.

Response: Thanks for the question. We designed new experiment to illustrate this. We IPed endogenous STX17 and determined the IPs with endogenous SNAP29 and VAMP8 under ULK1 WT, S423A and S423D overexpression condition. The data was presented as **Fig. 5g**.

Figure 5

Figure 5. Phosphorylation of ULK1 influences STX17-SNAP29-VAMP8 complex. (g) The myc-ULK1 variants transfected HeLa cells were treated with CQ or PBS. The IP assay was performed with STX17 antibody or IgG. and detected by indicated antibodies. The IP bands were quantified. The IP bands were quantified. All values are means \pm SEM of at least three independent experiments. Student's t test (unpaired); *, $p < 0.05$, ** $p < 0.01$, *** $p < 0.001$.

2. In the absence of standard in vitro fusion experiments involving proteoliposomes reconstituted with SNAREs, it is hard to conclude the impact on the membrane fusion step. A failure of recruiting

STX17 to autophagosomes can be an alternative explanation for the accumulation of double-membrane structures.

Response: Thank you for your great suggestion. We would address these issues by experiments and discussion.

- 1) We use a variety of experimental techniques to prove the issue in several ways indirectly (**Fig.4 and Fig.5**). For instance, Immunofluorescence experiments indicate that STX17 and LC3 are better colocalized in the presence of S423A (**Fig.5c**). It indicates that STX17 is more likely recruited to the autophagosome by S423A. Similarly, endogenous CO-IP experiments showed that, in the presence of S423A, STX17 has a strong ability to interact with Snap29 and VAMP8 (**Fig.5g**). Therefore, it is suggested that S423A enhances the STX17-mediated fusion step of autophagy. In conclusion: ULK1s do not prevent the formation of autolysosome by binding to STX17.
- 2) We used another reliable technology to demonstrate our conclusion instead of in vitro fusion experiments. Proximity ligation assay (in situ PLA) is a technology that extends the capabilities of traditional immunoassays to include direct detection of proteins, protein interactions and modifications with high specificity and sensitivity¹². Protein targets can be readily detected and localized with single molecule resolution and objectively quantified in unmodified cells and tissues. Utilizing only a few cells, sub-cellular events, even transient interactions, are revealed in situ and sub-populations of cells can be differentiated. We used both endogenous conditions and PLA system to confirm the interaction between ULK1 and STX17 interaction. The endogenous binding assay is shown in Fig. 4b and Fig.4c. We also performed PLA assay for different conditions such as serum starvation or ULK1 mimics in normal condition. All of these PLA related data are **Fig.4h, Fig.4i**, and **Fig.4j**. Thanks for your understanding.

We do observe the double membrane structure accumulation is in S423D re-expressed in ULK1/2 DKO cells (**Fig. 3c, S4c**). The STX17 is still colocalized with LC3 in S423D cells (**Fig. 5c**). It

indicates that STX17 is still recruited to autophagosome in these cells to a certain degree. So in our study the double membrane structure accumulation is not due to a failure of recruiting STX17 to autophagosomes. Again, thank you for your great question.

Figure 5

Figure 5. Phosphorylation of ULK1 influences STX17-SNAP29-VAMP8 complex. (c,d) GFP-STX17, Flag-LC3 and Myc-ULK1 variants were cotransfected and detected by immunostaining in HeLa cells (n=20) and the quantification of images (d). Scale bars, 10 μ M. All values are means \pm SEM of at least three independent experiments. Student's t test (unpaired); ***p< 0.001. (g) The myc-ULK1 variants transfected HeLa cells were treated with CQ or PBS. The IP assay was performed with STX17 antibody or IgG. and detected by indicated antibodies. The IP bands were quantified. The IP bands were quantified. All values are means \pm SEM of at least three independent experiments. Student's t test (unpaired); *,p<0.05, **p< 0.01, ***p<0.001.

Figure 4

Figure 4. ULK1 physically interacts with STX17. (h) Myc-ULK1 or Myc vector were cotransfected with GFP-STX17. The PLA assay was performed and the signals (red) were detected under Confocal Microscopy. Scale Bar, 10 μ M. (i) The HeLa cells were transfected with GFP-STX17 and Myc-ULK1 with/out serum starvation (S.S). The PLA assay was performed with Myc and GFP antibodies. The fluorescence signal (red) was observed under Confocal Microscopy. Scale Bar, 10 μ M. Average of signal was quantified with a minimum of 20 cells. All values are means \pm SEM of at least three independent experiments. Student's t test (unpaired); *** p < 0.001. (j) The representative images of PLA assay of GFP-STX17 cotransfected with myc-ULK1 WT, S423A, S423D or myc-vector. Scale Bar, 10 μ M. Average of signal was quantified with a minimum of 20 cells. All values are means \pm SEM of at least three independent experiments. Student's t test (unpaired); ** p < 0.01, *** p <0.001.

Figure 3

Figure 3. Phosphorylation of ULK1 play a key role in fusion of autophagosome to lysosome. (c) Representative confocal images of ULK-DKO HeLa cells transfected with Myc-ULK1 WT and mutants shown as the average of puncta per cell. Scale Bar, 10 μ M. A minimum of 20 cells was counted. All values are means \pm SEM of at least three independent experiments. Student's t test (unpaired); *** p < 0.001 ns: No significance. **(d)** The quantificated ratio(CQ-ULK1s/ULK1s) of puncta per cell were shown. All values are means \pm SEM of at least three independent experiments. Student's t test (unpaired); *** p < 0.001.

Figure S4

Supplemental Figure 4. Phosphorylation of ULK1 regulates autophagosome fusion. (c) Accumulation of complete double membrane autophagosome (red star) by TEM in ULK1 mutants stable cells in ULK1-KO cells with/without CQ treatment (n=15). N: Nuclei.

3. All results to support “ULK1 can be phosphorylated by PKC α at Serine 423 in vivo and in vitro” were done by overexpression or Co-IP in the cell. There is no direct evidence to demonstrate the interaction between ULK1 and PKC α . As shown in Fig2f, the inhibition of PKC α can also come from reducing the mTORC activity, which should be excluded through appropriate control experiments.

Response: Thank you for your great suggestion.

For your concern, we used purified ULK1 truncation from bacterial and purified PKC α from insect cells for pulldown assay(**Fig.S2c**). We show that there is a direct interaction between the two proteins. Another Reliable evidence is that in vitro kinase assay using recombinant PKC α and ULK1 fragments to demonstrate the direct phosphorylation via Ala scanning technology and LC-MS/MS (**Fig.1c,S1b**), line 112-121, page 4-5.

For your another concern, in order to demonstrate relationship of the activity of PKC and the activity of mTOR, we designed new experiments. We knockdown Raptor or mTOR in HEK293 and Hela cells, then we detected the PKC activities and quantified these data. Knockdown of Raptor or mTOR decrease PKC activities (**Fig.S3c,S3d**). Similarly, knockdown of PKC α in these two cells also leads to the reduction of mTORC activity (**Fig. S3e**). These

results indicated that PKC α and mTORC1 activities could affect each other.

Thank you again for your great suggestion.

Figure S2

Supplemental Figure 2. PKC α physically interacts with ULK1. (c) Purified GST or GST-ULK1 279-525 were incubated with His-PKC α , His-PKC α is purified from Sf9 insect cells. After washing, the pulldown result was analyzed with His antibody.

Figure 1

Figure 1. S423 of ULK1 is the phosphorylation site by PKC α . (c) Mapping phosphorylation site of ULK1 by in vitro kinase assay.

Figure S1

Fig. S1 PKC phosphorylates ULK1 in vitro and in vivo. (b) After the in vitro PKC α kinase assay, the ULK1 279-525 was pulled down by GST beads, washed for 3 times by PBS, and followed by PRM of LC-MS/MS to

identify the accurate phosphorylation site.

Figure S3

Supplemental Figure 3. PKC regulating autophagy is related to mTORC1.

(c,d) The PKC α and mTORC1 activities were determined by p-PKC substrate and p-4EBP antibodies after Raptor or mTOR was knocked down in both HeLa and HEK293 cells. The PKC activities were quantified. All values are means \pm SEM of at least three independent experiments. Student's t test (unpaired); ** $p < 0.01$, *** $p < 0.001$. (e) The PKC α and mTORC1 activities were determined by p-PKC substrate and p-mTOR antibodies after PKC α was knocked down in HeLa. The mTOR activities were quantified. All values are means \pm SEM of at least three independent experiments. Student's t test (unpaired); * $p < 0.05$.

Finally, this manuscript is not well prepared. Typos and fragments gave me a hard time to read. I was also frustrated by the low quality of images and poor arrangement of figure panels. For example, based on the poor resolution of Figure 3c, it's not easy for one to claim "accumulation of completed double membrane autophagosomes".

Response: Thank you for your great advice. As suggested we proofread carefully and modified the maintext and figures.

1. Al-Dhaheri MH, Rowan BG. Application of phosphorylation site-specific antibodies to measure nuclear receptor signaling: characterization of novel phosphoantibodies for estrogen receptor alpha. *Nuclear receptor signaling* **4**, (2006).
2. Yan X, *et al.* Yin Yang 1 (YY1) synergizes with Smad7 to inhibit TGF- β signaling in the nucleus. *Science China Life sciences* **57**, 128-136 (2014).
3. Liang J, *et al.* Mitochondrial PKM2 regulates oxidative stress-induced apoptosis by stabilizing Bcl2. *Cell research* **27**, 329-351 (2017).
4. Dang F, *et al.* Insulin post-transcriptionally modulates Bmal1 protein to affect the hepatic circadian clock. *Nature communications* **7**, 12696 (2016).
5. Diao J, *et al.* ATG14 promotes membrane tethering and fusion of autophagosomes to endolysosomes. *Nature* **520**, 563-566 (2015).
6. Yang M, *et al.* A C9ORF72/SMCR8-containing complex regulates ULK1 and plays a dual role in autophagy. *Science advances* **2**, (2016).
7. Itakura E, Kishi-Itakura C, Mizushima N. The hairpin-type tail-anchored SNARE syntaxin 17 targets to autophagosomes for fusion with endosomes/lysosomes. *Cell* **151**, 1256-1269 (2012).
8. Itakura E, Mizushima N. Syntaxin 17: the autophagosomal SNARE. *Autophagy* **9**, 917-919 (2013).
9. Lv L, *et al.* Acetylation targets the M2 isoform of pyruvate kinase for degradation through chaperone-mediated autophagy and promotes tumor growth. *Molecular cell* **42**, 719-730 (2011).
10. Yu L, *et al.* Termination of autophagy and reformation of lysosomes regulated by mTOR. *Nature* **465**, 942-946 (2010).
11. Zhang T, *et al.* G-protein-coupled receptors regulate autophagy by ZBTB16-mediated ubiquitination and proteasomal degradation of Atg14L. *eLife* **4**, (2015).
12. Gullberg M, *et al.* Cytokine detection by antibody-based proximity ligation. *Proc Natl Acad Sci U S A* **101**, 8420-8424 (2004).

Reviewer #1 (Remarks to the Author):

Authors have done a huge effort to improve the manuscript, most of the missing information, experiments, quantifications and controls are now provided by authors in this new version. Overall the manuscript is now appropriate for publication; however there are still many details and missing information regarding the methodology and the conditions used in each experiment that is not provided by authors. The limit in the number of characters cannot be used to justify the lack of information in most of the experiments and in material and methods. Figure 6i is a good example to illustrate this concern: what is PMA? And CHX? What are these drugs used for? What are the conditions (concentration and time) used? What means L.E. and S.E.? And Myc? Is ULK1 tagged to Myc and overexpressed in these conditions? How? All this information is totally missing in the text, in the figure legend and in Materials & Methods. All information provided by authors regarding this experiment is "ULK1 degradation is much slower in LAMP2-KO cell than in control cells under PMA treatment". As a reader it is not possible to figure out the details of the experiment and obviously is not possible to reproduce the experiment. This example could be extrapolated to almost all the experiments in the manuscript.

Reviewer #2 (Remarks to the Author):

In the revised version of the manuscript by Wang et al., the authors investigated the role of ULK1 for the fusion of autophagosomes with lysosomes. I still think that the authors make an important observation which might clarify the role of ULK1 during later steps of autophagy. I appreciate that the authors performed several additional experiments. However, the arrangement of the entire manuscript is somewhat confusing. There are still several mistakes within the text of the manuscript, and the overall quality of the English language is low. I think these issues need careful revision, perhaps supported by the editorial office. Generally, I think this study is of interest for the readership of NATURE COMMUNICATIONS. The following aspects need to be addressed (I am referring to my previous numbering):

Major points:

1)

a. The authors state the WT ULK1 might be degraded through both proteasome and lysosome. Can the degradation of WT ULK1 be blocked by the simultaneous usage of MG132 and CQ?

b. Here the authors just state that there is crosstalk between PKC and mTOR. However, I can still not understand how the phospho-status of ULK1 at Ser423 is regulated during autophagy.

c. OK

d. This question aimed at the oscillation of phospho-Ser423 shown in the new figure 1J. The authors state that reactivation of PKC is possible. Perhaps they can include the phospho-PKC substrate antibody used in other figures to support their conclusion.

2) The authors quantified the former figure 5G (new figure S7B). In my view, the quantification is not supported by the accompanying immunoblot, but I will not insist on this point. Still I think there is a lack of mechanistic insight how non-phosphorylated ULK1 regulates the interaction between STX17 and SNAP29.

3) Ok

Minor points:

1) Still the manuscript needs considerable proofreading. The authors state that they proofread the manuscript, but this I strongly doubt. Few examples are (but there are several more):

a. Lines 140-144: the authors state that U87 cells were used. In the legend of figure 1H it is stated that HeLa cells were used

b. Line 228: ULK1 knockout ULK2 knockdown cells should not be termed ULK12-DKO cells

c. Figure S2: does the quantification shown in panel H belong to the blot shown in E?

d. Figure S2 (legend): HA-PKC and FLAG-ULK1 were used, and the pulldown was done with glutathione beads???

- e. Figure S3A: PKC inhibitors alone are not shown; additionally, it does not make sense to calculate significance between combinatorial treatments and DMSO
 - f. Figure S4B: is the labeling Ctr. and ULK1-KO correct?
 - g. Figure S6D-F: there is no labeling of the x-axis
 - h. There are several spelling errors in the text.
- 2)
- a. The authors state that the increased ATG13 phosphorylation induced by the PKC inhibitor is caused by the effect of the PKC inhibitor on mTOR; this might be possible, but I think it is difficult to draw a conclusion from this experiment.
 - b. OK
 - c. OK (but please consider my major comment 1A)
- 3) Ok (but please consider my minor comment 1B).
- 4) OK
 - 5) OK
 - 6) Can the authors explain the p-S423 antibody was not used for step 2?
 - 7) OK
 - 8) OK
 - 9) OK
 - 10) OK
 - 11) OK
 - 12) I doubt that the description of the MS experiments meets the criteria for publication in NATURE COMMUNICATIONS.
 - 13) The authors should rephrase their observation; at present, the description of Figure S6F cannot be understood.

Finally, I strongly encourage the authors to remove redundant experiments and to optimize the presentation of the panels within individual figures.

Reviewer #3 (Remarks to the Author):

In the revised version, authors significantly improved data quality. However, claims of this manuscript are still mainly based on overexpression of tag-fused proteins followed by immunoprecipitation or pulldown. The golden standard involving in vitro proteoliposomes to check fusogenic activity is missing for "fusion-enhancing activity." Previously, it has been demonstrated that the assembly of SNARE core domain is not enough to drive full fusion of two membranes [see Figure 4 in Hernandez et al Science 336, 1581, 2012]. Therefore, there is no direct correlation between protein pulldown experiments and membrane fusion measurements. One cannot use in situ PLA to "to demonstrate our conclusion instead of in vitro fusion experiments." As stated in the rebuttal letter, "Proximity ligation assay (in situ PLA) is a technology that extends the capabilities of traditional immunoassays to include direct detection of proteins, protein interactions and modifications with high specificity and sensitivity". Alternatively, a cell-cell fusion experiment involving fluorescent proteins can be performed to support the impact of ULK1 on membrane fusion.

Response to Reviewers

Reviewer #1 (Remarks to the Author):

Authors have done a huge effort to improve the manuscript, most of the missing information, experiments, quantifications and controls are now provided by authors in this new version. Overall the manuscript is now appropriate for publication; however there are still many details and missing information regarding the methodology and the conditions used in each experiment that is not provided by authors. The limit in the number of characters cannot be used to justify the lack of information in most of the experiments and in material and methods. Figure 6i is a good example to illustrate this concern: what is PMA? And CHX? What are these drugs used for? What are the conditions (concentration and time) used? What means L.E. and S.E.? And Myc? Is ULK1 tagged to Myc and overexpressed in these conditions? How? All this information is totally missing in the text, in the figure legend and in Materials & Methods. All information provided by authors regarding this experiment is "ULK1 degradation is much slower in LAMP2-KO cell than in control cells under PMA treatment". As a reader it is not possible to figure out the details of the experiment and obviously is not possible to reproduce the experiment. This example could be extrapolated to almost all the experiments in the manuscript.

Response: Thanks for your great concern. As suggested we proofread carefully and added the details in main text and Figure Legend including Figure 6i. To make the reader easily understand our description We also modified "ULK1 degradation is much slower in LAMP2-KO cell than in control cells under PMA treatment" to "PMA treatment triggers ULK1 degradation faster in WT cells than in LAMP2-KO cells" in main text from line 431 to 432 line .

Reviewer #2 (Remarks to the Author):

In the revised version of the manuscript by Wang et al., the authors investigated the role of ULK1 for the fusion of autophagosomes with lysosomes. I still think that the authors make an important observation which might clarify the role of ULK1 during later steps of autophagy. I appreciate that the authors performed several additional experiments. However, the arrangement of the entire manuscript is somewhat confusing. There are still several mistakes within the text of the manuscript, and the overall quality of the English language is low. I think these issues need careful revision, perhaps supported by the editorial office. Generally, I think this study is of interest for the readership of **NATURE COMMUNICATIONS**. The following aspects need to be addressed (I am referring to my previous numbering):

Response: Thank you for your affirmation of our observation and concern about the English. We ask Dr. Zachary Ende (Emory University), a native English speaker, to polish the English for us. We appreciate Dr. Ende for the English editing.

Major points:

1)

a. The authors state the WT ULK1 might be degraded through both proteasome and lysosome. Can the degradation of WT ULK1 be blocked by the simultaneous usage of MG132 and CQ?

Response: Thanks for your questionnaire. Most proteins are degraded through either proteasome or lysosome. Also it reported that many proteins are degraded through both proteasome and lysosome in literature. In this manuscript we state that ULK1 WT may be degraded through both and lysosome too. As the reviewer suggested, we analyzed the ULK1 after the cells were treated with MG132 and CQ or DMSO as negative control. This result is in the Fig. S8c and showed simultaneous usage of MG132 and CQ does block the ULK1 WT degradation.

b. Here the authors just state that there is crosstalk between PKC and mTOR. However, I can still not understand how the phospho-status of ULK1 at Ser423 is regulated during autophagy.

Response: In our data ULK1 S423 is phosphorylated by PKC-alpha and this phosphorylation displays oscillation during autophagy. Because of the activity of PKC is positively related to mTORC1 and this phosphorylation(p-S423) is also displays the positive relationship to mTORC1 activity. Both p-S757 (By mTOR) and p-S423 (By PKC-alpha) of ULK1 are phosphorylated under nutrient rich conditions. Even p-S423 site shows the same pattern with mTORC1 activity but it's not a substrate of mTORC1. Normally, mTORC1 recognize T/S-P, however, SXR is in the ULK1 of S423. This is the canonical phosphorylation site of PKC.

In a word, inhibiting autophagy by active kinases(mTOR, PKC-alpha) under nutrient rich conditions is achieved by exerting the retardance at different stages of autophagy respectively and simultaneously. MTOR inhibits the early stage of autophagy, and PKC-Alpha inhibits the late stages of autophagy. We also added the statement in discussion from line 481 to line 485. Thanks for your great concern.

c. OK

d. This question aimed at the oscillation of phospho-Ser423 shown in the new figure 1J. The authors state that reactivation of PKC is possible. Perhaps they can include the phospho-PKC substrate antibody used in other figures to support their conclusion.

Response: Thanks for your suggestion. We used the phospho-PKC substrate antibody to detect the whole cell lysates in fig.S2h. we also probed the samples of Fig. 5a with phospho-PKC substrate antibody and the panels were inserted into the figure.

2) The authors quantified the former figure 5G (new figure S7B). In my view, the quantification is not supported by the accompanying immunoblot, but I will not insist on this point. Still I think there is a lack of mechanistic insight how non-phosphorylated ULK1 regulates the interaction between STX17 and SNAP29.

Response: Thanks for your understanding and great concern. ULK1 is a serine/threonine-specific protein kinase. Therefore, it is possible that ULK1 can act as a kinase to play a role in the later stage of autophagy.

We deduced that ULK1 could act as a kinase and phosphorylate STX17, consequently the affinity between STX17 and Snap29, VAMP8 will be attenuated or enhanced after STX17's phosphorylation, then, the late stage of autophagy process will be changed, This change depends on the regulation of STX17 by the S423-phosphorylation state of ULK1.

The significant of the involvement of ULK1 in the later stages of autophagy is self-evident, and its function is very crucial. If the hypothesis(ULK1 is a kinase for STX17) is correct, our study will be more valuable. At first we were willing to think it wise to go down this line of thought.

We did our biggest effort including time (about 1 year) and large sums of money to demonstrate that ULK1 is a kinase for STX17. For example, we used ULK1 kinase assay to perform pho-tag assay and Isotope experiment (data not show) to demonstrate the phosphorylation of STX17 by ULK1. We also cooperated with other labs using mass spectrometry to search the phosphorylation of STX17 for many times, but we did not find the evidence. Besides, knases do not exert kinase activity even its binding to other proteins, which is a common phenomenon. For example, IRE1-alpha, a kinase, does not show any kinase activity to its binding proteins(except JNK and itself).

Therefore, based on our experiments, ULK1 is not a kinase for STX17; ULK1 regulates the late stage of autophagy by transient protein-protein interaction.

It is a common mechanism of protein-Protein regulation that "A" protein binding to "B" Protein to promote the interaction between "B" protein and "C" protein^{1, 2}. For example, HSC90 α 1 promotes the binding of PKM2 to Bcl2².

Our model is: STX17 and Snap29 participate in the fusion of autophagosome and lysosome, We found that the interaction between STX17 and Snap29 is mediated by ULK1. S423A or non-P-S423-ULK1 enhanced the STX17 and Snap29's interaction (Fig.4g, Fig.5a,b,g,h). STX17 and Snap29's interaction requires the binding between ULK1 and STX17. Overexpression of Snap29 decrease the binding between ULK1 and STX17 (Fig.S7b. Knockdown of Snap29 increase the binding between ULK1 and STX17 (Fig.S7c) . In conclusion, after enhancement of STX17 and Snap29' s interaction, non-p-ULK1 or S423A is dissociated from STX17 and degraded through preteasome pathway (Fig.6a,b). The formation of autolysosome is affected by transient interaction between non-p-S423-ULK1 and STX17.

Again, Thanks for your great concern and understanding.

3) Ok

Minor points:

- 1) Still the manuscript needs considerable proofreading. The authors state that they proofread the manuscript, but this I strongly doubt. Few examples are (but there are several more):
 - a. Lines 140-144: the authors state that U87 cells were used. In the legend of figure 1H it is stated that HeLa cells were used

Response: Thank you for reminding us of this mistake. We modified the "HeLa cells" to "U87 cells" in the legend of Figure 1H.

b. Line 228: ULK1 knockout ULK2 knockdown cells should not be termed ULK12-DKO cells

Response: As suggested, we named ULK1 knockout and ULK2 knockdown cells as ULK1/2-KO/D and changed all of them through the manuscript.

c. Figure S2: does the quantification shown in panel H belong to the blot shown in E?

Response: It's mistake of labeling. There is no figure S2H in previous version. We delete the "h" in the figure S2. Thank you for reminding us of this mistake.

d. Figure S2 (legend): HA-PKC and FLAG-ULK1 were used, and the pulldown was done with glutathione beads???

Response: We made a mistake here. We correct this mistake in figure s2e. We also deleted the previous Figure s2h legend. Thank you for reminding us of this mistake.

e. Figure S3A: PKC inhibitors alone are not shown; additionally, it does not make sense to calculate significance between combinatorial treatments and DMSO

Response: Thanks for your suggestion. We added the PKC inhibitors alone : Bis I treatment in fig. S3a. We also recalculated the significance base on current results.

f. Figure S4B: is the labeling Ctr. and ULK1-KO correct?

Response: Thanks for reminding us of this mistake. We changed "Ctr." to "WT" and "ULK1-KO" to "ULK1/2-KO/D".

g. Figure S6D-F: there is no labeling of the x-axis

Response: Thanks for your concern. We added the labeling of x-axis .

h. There are several spelling errors in the text.

Response: We screened the article carefully and corrected many mistakes, thanks.

2)

a. The authors state that the increased ATG13 phosphorylation induced by the PKC inhibitor is caused by the effect of the PKC inhibitor on mTOR; this might be possible, but I think it is difficult to draw a conclusion from this experiment.

Response: Thanks for your concern. We have improved the statement in the article: " These results may be due to a reduction in mTORC1 activity" . Please see it in main text line 213.

b. OK

c. OK (but please consider my major comment 1A)

3) Ok (but please consider my minor comment 1B).

4) OK

5) OK

6) Can the authors explain the p-S423 antibody was not used for step 2?

Response: Thank you for your question. 1) As to Step 2, the in vitro PKC-alpha kinase assay for P-S423 has not been carried out; 2) All of the phosphorylations were wiped out in the CIP process in step 2. By combining the above two points, we examined the Total-S/T.

We used P-423 antibody to detect the step 4 just because the CIP process in step 4 wiped out only phosphorylated S423 site, other phosphorylation sites is exterminated in step 2. Based on your concern, we also detected the P-423 of the samples of Fig. 1f via P-S423 specific antibody, the process of CIP for P-423 were also successful.

If requested by the reviewers, we can also include the data in Fig 1f. Thanks again for your understanding.

7) OK

8) OK

9) OK

10) OK

11) OK

12) I doubt that the description of the MS experiments meets the criteria for publication in **NATURE COMMUNICATIONS**. (MS)

Response: Thanks for your concern. We modified the description of the MS experiments in Material and Methods line 656 to line 659. Thanks for your understanding.

13) The authors should rephrase their observation; at present, the description of Figure S6F cannot be understood.

Response: Thanks for your suggestion. We rephrased our observation both in main text and Figure legend:

Fig. S6(d,e,f)The colocalization of Atg13, Atg16L, and LAMP2a with ULK1s were measured by Pearson correlation analysis and independent-sample t-tests. The represented as quantification from a,b,c. A minimum of 20 cells. All values are means \pm SEM of at least three independent experiments. Student's t test (unpaired); * $p < 0.05$, ** $p < 0.001$.

Finally, I strongly encourage the authors to remove redundant experiments and to optimize the presentation of the panels within individual figures.

Response: As suggested, we deleted several figure such as Figure S1c,S1d and figure S7a.

Reviewer #3 (Remarks to the Author):

In the revised version, authors significantly improved data quality. However, claims of this manuscript are still mainly based on overexpression of tag-fused proteins followed by immunoprecipitation or pulldown. The golden standard involving in vitro proteoliposomes to check fusogenic activity is missing for "fusion-enhancing activity." Previously, it has been demonstrated that the assembly of SNARE core domain is not enough to drive full fusion of two membranes [see Figure 4 in Hernandez et al Science 336, 1581, 2012]. Therefore, there is no direct correlation between protein pulldown experiments and membrane fusion measurements. One cannot use in situ PLA to "to demonstrate our conclusion instead of in vitro fusion experiments." As stated in the rebuttal letter, "Proximity ligation assay (in situ PLA) is a technology that extends the capabilities of

traditional immunoassays to include direct detection of proteins, protein interactions and modifications with high specificity and sensitivity". Alternatively, a cell-cell fusion experiment involving fluorescent proteins can be performed to support the impact of ULK1 on membrane fusion.

Response: Thanks for the critical concern, according to the suggestion, we improved the quality of the study greatly. As suggested we performed an in vitro fusion assay in the presence of purified GST-ULK1s. The procedure is described in Materials and Methods³ and the result as Figure 3e. From this experiment we observed that ULK1 S423D prevents membrane fusion in vitro.

1. Yan X, *et al.* Yin Yang 1 (YY1) synergizes with Smad7 to inhibit TGF- β signaling in the nucleus. *Science China Life sciences* **57**, 128-136 (2014).
2. Liang J, *et al.* Mitochondrial PKM2 regulates oxidative stress-induced apoptosis by stabilizing Bcl2. *Cell research* **27**, 329-351 (2017).
3. Moreau K, Puri C, Rubinsztein DC. Methods to analyze SNARE-dependent vesicular fusion events that regulate autophagosome biogenesis. *Methods (San Diego, Calif)* **75**, 19-24 (2015).

Reviewer #2 (Remarks to the Author):

In the re-revised version of the manuscript by Wang et al., the authors investigated the role of ULK1 for the fusion of autophagosomes with lysosomes. Once more I would like to point out that I am convinced that the authors make an important observation, and this work complements the understanding of ULK1 signaling. The quality of the English language has improved. However, the arrangement and the labeling of several figures are still incredibly confusing. This concern has now been raised several times, also by the other two reviewers. I really hope that this can be fixed by the editors, and I will list the most obvious examples. In my comments below, I am referring to my previous numbering.

Major points:

- 1)
 - a. OK
 - b. Although I can still not understand how PKC activity towards ULK1 is regulated during autophagy, I will not insist on this point. The upstream signaling machinery of MTOR is relatively well established, and I just wanted to know whether there is similar knowledge with regard to PKC (under proautophagic conditions).
 - c. OK
 - d. Ok
- 2) Still I think there is a lack of mechanistic insight how non-phosphorylated ULK1 regulates the interaction between STX17 and SNAP29. At least the authors should mention that ULK1 does not phosphorylate STX17 and that ULK1 exerts its effect on autolysosome formation through transient protein-protein interaction between non-p-S423-ULK1 and STX17.
- 3) Ok

Minor points:

- 1) Although the quality of the English language has improved, I think there is still a lot of work for the editors.
 - a. OK
 - b. OK
 - c. OK
 - d. OK
 - e. Now there are two blots in Figure S3A, and some treatments are identical for the two blots. The quantifications belong to the new blot? This figure should be carefully revised, and one blot should be shown
 - f. Figure S4B: autophagosomes are not labeled by a red star but by a red dot
 - g. OK
 - h. OK
- 2)
 - a. OK
 - b. OK
 - c. OK
- 3) Ok
- 4) OK
- 5) OK
- 6) As suggested by the authors, the p-423 blot of step 2 should be included in figure 1F.
- 7) OK
- 8) OK
- 9) OK
- 10) OK
- 11) OK

12) Still, the description of the MS experiments is insufficient (exact preparation of samples, used mass spectrometer, software used for peptide identification, etc.).

13) Still, the figure legend to S6D,E,F is difficult to understand. Example: "The represented as quantification from a,b,c. A minimum of 20 cells."

Finally, I want to emphasize that the presentation of the figures/panels is still of poor quality. Few examples are given below. This point has now been raised by the other reviewers and me, and I cannot understand that I have to read this manuscript now for the third time without any improvement. I think the quality of figure presentation does not warrant publication in Nature communications.

Examples:

Figure 1: the panels are arranged randomly in order to fit to one page; in figure 1F is should be "kinase assay" for step 3.

Figure 2D/E: It is not easy to see which quantification belongs to which blot. The bands in figure 1E are fainter compared to other western blots in the manuscript.

Figure 3E: "The fusion percentage were quantified". How?

Figure 4B: it should be IgG instead of IGg

Figure 4D/E: in D different STX17 truncations are explained. In panel E it should be stated that ULK1 truncations are depicted.

Figure 4I/J: The presentation of quantifications has to be optimized (perhaps always on the same side of the fluorescence images)

Figure 5A,C,D: Again the presentation of the quantifications is confusing.

Figure 5F: Usually the authors made use of chloroquine. Why Baf A1 was used in this figure?

Figure S2F: It should be indicated that a GST pulldown was performed

Figure S2G: It should be indicated that ULK1 variants are depicted.

Figure S2H: Time points are missing.

Figure S2I: Were 2 mice analyzed? This should be indicated somewhere.

Figure S3A: As mentioned above, two blots are shown. Quantifications obviously belong to the right blot. This panel has to be optimized.

Figure S3E: In figures S3C and S3D, p-4EBP was used as readout for mTOR activity. This should also be done for figure S3E.

Figure S4B: As mentioned above, there are no red stars but red dots.

Figure S7: Again, quantifications should be depicted on the same side of the blots.

Reviewer #3 (Remarks to the Author):

My major concern about fusion has been addressed.

REVIEWERS' COMMENTS:

Reviewer #2 (Remarks to the Author):

In the re-revised version of the manuscript by Wang et al., the authors investigated the role of ULK1 for the fusion of autophagosomes with lysosomes. Once more I would like to point out that I am convinced that the authors make an important observation, and this work complements the understanding of ULK1 signaling. The quality of the English language has improved. However, the arrangement and the labeling of several figures are still incredibly confusing. This concern has now been raised several times, also by the other two reviewers. I really hope that this can be fixed by the editors, and I will list the most obvious examples. In my comments below, I am referring to my previous numbering.

Major points:

1)

a. OK

b. Although I can still not understand how PKC activity towards ULK1 is regulated during autophagy, I will not insist on this point. The upstream signaling machinery of MTOR is relatively well established, and I just wanted to know whether there is similar knowledge with regard to PKC (under proautophagic conditions).

Response: Under rich nutritional conditions, autophagy process is inhibited^{1, 2}. Inhibiting autophagy by active kinases (mTOR, PKC) under nutrient rich conditions is achieved by exerting the retardance at different stages of autophagy respectively and simultaneously. mTOR inhibits the early stage of autophagy, and PKC-Alpha inhibits the late stages of autophagy as showed in (Supplementary Figure 10) if nutritional signaling comes, mTOR is activated and phosphorylate S757 of ULK1 and many Beclin1-Vps34 complex proteins to inhibit the initiation steps of autophagy^{2, 3}. PKC inhibit late autophagy steps through phosphorylation of LC34 and ULK1 (our study).

We have demonstrated that the activity of mTOR and PKC are linked. their activity is affected

by each other, if one of them is activated, the other one's activity is also enhanced. So they cooperate to inhibit the whole process of autophagy respectively and simultaneously.

c. OK

d. Ok

2) Still I think there is a lack of mechanistic insight how non-phosphorylated ULK1 regulates the interaction between STX17 and SNAP29. At least the authors should mention that ULK1 does not phosphorylate STX17 and that ULK1 exerts its effect on autolysosome formation through transient protein-protein interaction between non-p-S423-ULK1 and STX17.

Response: As requested, we discuss this in the Discussion from line 476 to line 486.

3) Ok

Minor points:

1) Although the quality of the English language has improved, I think there is still a lot of work for the editors.

Response: We asked the professional English editing service suggested by Nature Communication editors. We also upload the revised version with the edit tracking as a file.

a. OK

b. OK

c. OK

d. OK

e. Now there are two blots in Figure S3A, and some treatments are identical for the two blots. The quantifications belong to the new blot? This figure should be carefully revised, and one blot should be shown

Response: As suggested, we keep one blot and its quantification chart in Supplementary Figure 3a. Additionally, we describe the quantification chart clearly for which blot in the Figure Legend.

f. Figure S4B: autophagosomes are not labeled by a red star but by a red dot

Response: We relabeled the autophagosome with the RED STAR in Figure S4B. Thank you for the correction.

g. OK

h. OK

2)

a. OK

b. OK

c. OK

3) Ok

4) OK

5) OK

6) As suggested by the authors, the p-423 blot of step 2 should be included in figure 1F.

Response: We included the p-423 blot of step 2 in the Figure 1F.

7) OK

8) OK

9) OK

10) OK

11) OK

12) Still, the description of the MS experiments is insufficient (exact preparation of samples, used mass spectrometer, software used for peptide identification, etc.).

Response: Thanks for the suggestion. We describe the detail of the MS experiments in the Material and Methods section including sample preparation, used software for peptide identification.

13) Still, the figure legend to S6D,E,F is difficult to understand. Example: “The represented as quantification from a,b,c. A minimum of 20 cells.”

Response: As requested, we modified the Figure Legend of Figure S6 as bellow to help the reader understand.

“The (d) represented as quantification from (a). The (e) represented as quantification from (b). The (f) represented as quantification from (c). A minimum of 20 cells.”

Finally, I want to emphasize that the presentation of the figures/panels is still of poor quality. Few examples are given below. This point has now been raised by the other reviewers and me, and I cannot understand that I have to read this manuscript now for the third time without any improvement. I think the quality of figure presentation does not warrant publication in Nature communications.

Examples:

Figure 1: the panels are arranged randomly in order to fit to one page; in figure 1F is should be “kinase assay” for step 3.

Response: As suggested, we rearranged the figures and panels and clearly indicated the quantification panel in the Legend. We modified the “Assay After CIP” to “Kinase Assay” for step3 in Figure 1F. We also add the p-S423 blot in the step3 in Figure 1F.

Figure 2D/E: It is not easy to see which quantification belongs to which blot. The bands in figure 1E are fainter compared to other western blots in the manuscript.

Response: We rearranged the quantification of Figure 2D to the right panel of the Figure 2D and described in the figure legend. The bands in Figure 1E are fainter than other blot but it is clear enough to see the difference of the phosphorylation between WT and S423A mutant.

Figure 3E: “The fusion percentage were quantified”. How?

Response: We add how to calculate the fusion percentage in the figure lengd of Figure 3E as “The fusion percentage were quantified by comparisons of the numbers of yellow dots to the numbers of red dots and yellow dots.”

Figure 4B: it should be IgG instead of Igg

Response: We correct the Igg to IgG.

Figure 4D/E: in D different STX17 truncations are explained. In panel E it should be stated that ULK1 truncations are depicted.

Response: As suggested we explained the GST-ULK1 truncations in Figure 4E.

Figure 4I/J: The presentation of quantifications has to be optimized (perhaps always on the same side of the fluorescence images)

Response: Because of space we can' t present the quantifications on the same side of the fluorescence images. But we indicated the each quantification clear in the figure legend.

Figure 5A,C,D: Again the presentation of the quantifications is confusing.

Response: We indicated the quantification in the Figure Legend.

Figure 5F: Usually the authors made use of chloroquine. Why Baf A1 was used in this figure?

Response: Similar with chloroquine, Baf A1 is also a inhibitor for lysosome activity. We want to use different drug to demonstrate one kind of thing, in our results, Baf A1 is also suitable. So we conclude this drug in our study.

Figure S2F: It should be indicated that a GST pulldown was performed

Response: As suggested, we indicated that “a GST pulldown was performed” in the figure legend of Figure S2F.

Figure S2G: It should be indicated that ULK1 variants are depicted.

Response: We explained the ULK1 fragments in the Figure legend and indicated the ULK1 variants are the same as in the Figure 4E.

Figure S2H: Time points are missing.

Response: We added the time points in the Figure S2H.

Figure S2I: Were 2 mice analyzed? This should be indicated somewhere.

Response: There are 3 mice analyzed each group. We indicated in the Figure Legend of Figure S2I.

Figure S3A: As mentioned above, two blots are shown. Quantifications obviously belong to the right blot. This panel has to be optimized.

Response: As suggested we deleted one blot and its quantification as in the minor point le.

Figure S3E: In figures S3C and S3D, p-4EBP was used as readout for mTOR activity. This should also be done for figure S3E.

Response: As suggested, we used the p-4EBP as the readout for mTOR activity in figure S3E.

Figure S4B: As mentioned above, there are no red stars but red dots.

Response: We relabeled the autophagosome with the RED STAR in Figure S4B.

Figure S7: Again, quantifications should be depicted on the same side of the blots.

Response: As suggested we rearranged the quantifications on the right of the blots.

Reviewer #3 (Remarks to the Author):

My major concern about fusion has been addressed.

1. He, C. & Klionsky, D.J. Regulation mechanisms and signaling pathways of autophagy. *Annual review of genetics* **43** (2009).
2. Klionsky, D.J. *et al.* Guidelines for the use and interpretation of assays for monitoring autophagy (3rd edition). *Autophagy* **12**, 1-222 (2016).
3. Kim, J., Kundu, M., Viollet, B. & cell biology, G.-K.L. AMPK and mTOR regulate autophagy through direct phosphorylation of Ulk1. *Nature cell biology* (2011).
4. Jiang, H., Cheng, D., Liu, W., Peng, J. & and biophysical, F.-J. Protein kinase C inhibits autophagy and phosphorylates LC3. *Biochemical and biophysical ...* (2010).